# Antimony oxide buffer layer for single- and double-junction perovskite-based solar cells

Biao Shi[1,2,3,4,5,9], Zetong Sunli[1,2,3,4,5,9], Pengfei Liu[1,2,3,4,5], Wei Han[1,2,3,4,5],
Rui Kong[1,2,3,4,5], Cong Sun[1,2,3,4,5], Ying Liu[1,2,3,4,5], Yuan Luo[1,2,3,4,5],
XianZhao Wang[1,2,3,4,5], Zhi Zhang[1,2,3,4,5], Dekun Zhang[1,2,3,4,5], Xiaona Du[1,2,3,4,5],
Fu Zhang[6], Miao Yang[6], Yongcai He [6] ✉, Bo He[6], Xixiang Xu [6], Rui Xia[7],
Xueling Zhang[7], Yifeng Chen [7], Jifan Gao [7] ✉, Fuzong Xu [8], Ying Zhao[1,2,3,4,5],
Stefaan De Wolf [8] ✉ & Xiaodan Zhang [1,2,3,4,5] ✉

Atomic layer-deposited tin oxide serves as an effective buffer layer in perovskite/silicon tandem solar cells due to its efficient charge extraction and sputtering tolerance. Nevertheless, its unavoidable chemical erosion effect of atomic layer-deposited tin oxide on perovskite requires thicker fullerene charge transport layers, leading to increased parasitic optical absorption. Herein, we firstly integrated thermal evaporated antimony oxide into solar cells to effectively replace atomic layer-deposited tin oxide, enabling a thinner fullerene to minimize optical losses and prevent damage to the perovskite. The unique amorphous-nanocrystalline structure of, antimony oxide facilitates ultrafast carrier transport via its embedded nanocrystalline network. The antimony oxide-based tandem solar cells demonstrated a significant improvement in power conversion efficiency compared to tin oxide-based devices, primarily due to an enhanced short-circuit current density of approximately $1\,mA/cm^2$ in the perovskite top cell. Remarkably, even at $64.64\,cm^2$ scale, the antimony oxide-based encapsulated large-area tandem solar cell retains an efficiency of 28.16% (with a certified value of 27.70%), attesting the scalability of this approach.

Monolithic perovskite/silicon tandem solar cells (PSTs) have emerged as highly promising photovoltaic devices, demonstrating exceptional power conversion efficiency (PCE) with a certified value of 34.85%[1]. Nevertheless, the considerable gap relative to the 42% theoretical limit points to critical unresolved losses in optical absorption and charge carrier transport[2]. The engineered functional layers in light-facing front contact architectures, featuring controlled nanoscale thickness[3], present significant potential for improving photovoltaic performance through enhanced short-circuit current density ($J_{SC}$).

Currently, thermal evaporated (TE) fullerene ($C_{60}$) and atomic layer-deposited (ALD) tin oxide ($SnO_X$) films are widely used as electron transport layer (ETL) and buffer layer, respectively, in nearly all p-

[1]Institute of Photoelectronic Thin Film Devices and Technology, Renewable Energy Conversion and Storage Center, State Key Laboratory of Photovoltaic Materials and Cells, Nankai University, Tianjin, P. R. China. [2]Tianjin Key Laboratory of Efficient Utilization of Solar Energy, Tianjin, P. R. China. [3]Haihe Laboratory of Sustainable Chemical Transformations, Tianjin, P. R. China. [4]Engineering Research Center of Thin Film Photoelectronic Technology of Ministry of Education, Tianjin, P. R. China. [5]Collaborative Innovation Center of Chemical Science and Engineering (Tianjin), Tianjin, P. R. China. [6]LONGi Central R&D Institute, LONGi Green Energy Technology Co. Ltd., Xi'an, P. R. China. [7]State Key Laboratory of PV Science and Technology, Trina Solar, Changzhou, P. R. China. [8]King Abdullah University of Science and Technology (KAUST), Physical Sciences and Engineering Division (PSE), Material Science and Engineering Program (MSE), Thuwal, Kingdom of Saudi Arabia. [9]These authors contributed equally: Biao Shi, Zetong Sunli. ✉e-mail: heyongcai@longi.com; Jifan.gao@trinasolar.com; stefaan.dewolf@kaust.edu.sa; xdzhang@nankai.edu.cn

i-n structured PSTs[4–13]. However, the intrinsic ligand exchange reaction between tetrakis(dimethylamino)tin(IV) (TDMASn) precursor in ALD and formamidinium (FA$^+$) in the perovskite is one of the main sources of the collapse of perovskite structure[14]. Therefore, it is essential to enhance the thickness and mechanical strength of $C_{60}$ layer to avoid adverse erosion dominated by ALD-$SnO_X$. However, this exacerbates the parasitic optical absorption at 300–550 nm wavelength which is a harbinger of the current loss in PSTs[15,16]. Besides, the additional ALD technology further complicates the tandem solar cell fabrication equipment. Thus, it is urgently desirable to develop a gentler technology to prepare the buffer layer to overcome the drawbacks of state-of-the-art $SnO_X$.

Herein, we firstly developed TE-antimony oxide ($Sb_2O_3$) film as an innovative alternative to ALD-$SnO_X$ for buffer layer applications. Through comprehensive characterization, we identified an amorphous-nanocrystalline hybrid structure in the $Sb_2O_3$ film and elucidated its unique electron transport mechanism, mediated by percolating nanocrystalline channels. Compared to ALD-$SnO_X$, TE-$Sb_2O_3$ demonstrates superior process window and minimized adverse effects on perovskite materials. This enhanced compatibility enables the reduction of the $C_{60}$ layer thickness from 15 to 5 nm without compromising perovskite device performance. Consequently, the $C_{60}$/$Sb_2O_3$ stacked functional layer achieves both excellent carrier extraction efficiency and minimized optical parasitic absorption losses. We employed a vacuum-solution hybrid deposition method to fabricate perovskite films with tunable bandgaps (1.59, 1.62, 1.64, and 1.68 eV) and integrated an ultra-thin $C_{60}$/$Sb_2O_3$ film into perovskite solar cells (PSCs), demonstrating universal bandgap applicability. The 1.64 eV-bandgap PSCs achieved a champion PCE of 23.18%, comparable to that of $SnO_X$-based PSCs (23.17%). Notably, the 23.18% PCE achieved here represents the highest reported value for mid-/wide-bandgap PSCs fabricated using a vacuum-solution hybrid method. Furthermore, $Sb_2O_3$-based PSTs exhibited an enhanced PCE of 30.28% with an aperture area of 1.0 cm², compared to the 28.59% efficiency of $SnO_X$-based PSTs, primarily due to a $J_{SC}$ improvement of ~1 mA/cm². The scalability of this approach was successfully demonstrated through the fabrication of 64.64 cm² encapsulated PSTs, which achieved a certified PCE of 27.70%—ranking among the highest reported efficiencies for larger-area PSTs (>10 cm²) in the literature.

## Results

### Antimony oxide thin film

Materials located at the junction of metallic and non-metallic elements in the periodic table are typically used as functional layers in semiconductor devices. N-type metal oxides, e.g. $SnO_X$ and titanium oxide ($TiO_X$)[17], have been widely applied in perovskite-based tandem solar cells. However, the high melting points of most metal oxides complicate their preparation via TE (Supplementary Table 1). In contrast, $Sb_2O_3$ is an n-type semiconductor with a low sublimation temperature (~490 °C under ambient pressure), owing to its molecular crystal structure[18]. Given this unique advantage, we proposed to prepare $Sb_2O_3$ film with evaporation method, as shown in Fig. 1a.

Atomic force microscope (AFM) image based on $Sb_2O_3$ shows a flat and homogenous morphology at the micrometer scale, without discernible voids or bumps (Supplementary Fig. 1). The morphology of 15 nm $Sb_2O_3$ coated on perovskite (PVK)/$C_{60}$ multilayer further validates its homogeneity and flatness by scanning electron microscope (SEM) and AFM (Supplementary Figs. 2, 3). However, only 5 nm $Sb_2O_3$ fails to fully cover the substrate (Supplementary Fig. 3). Transmission electron microscope (TEM) and X-ray diffraction (XRD) measurements indicate that the $Sb_2O_3$ forms an amorphous-nanocrystalline film[19,20] (Fig. 1b, c, Supplementary Fig. 4). The nanocrystalline structure was identified as cubic with an interplanar spacing of $d$ = 3.2 Å. The XRD data further reveals that the crystallites are preferentially oriented with the (222) planes parallel to the glass substrate[21,22].

To investigate the vertical conductivity of $Sb_2O_3$, we performed conductive atomic force microscopy (c-AFM), applying a 3 V bias to the longitudinal side of silicon/$Sb_2O_3$[23] (Fig. 1d–f). The c-AFM mapping clearly distinguishes between amorphous and nanocrystalline regions in the $Sb_2O_3$ layer (Fig. 1e). Our findings demonstrate that the nanocrystalline regions enable vertical conduction, whereas the amorphous regions remain insulation (Fig. 1f). Complementary lateral c-AFM measurements further reveal poor lateral conductivity in $Sb_2O_3$ in Supplementary Fig. 5. These results suggest that the nanocrystals can serve as conductive channels for longitudinal electrical transport, making the $Sb_2O_3$ potentially suitable as a buffer layer (Fig. 1g). However, as a wide-bandgap material (~4.25 eV), $Sb_2O_3$ tends to form high energy barriers at interface (Supplementary Fig. 6). We found that the presence of Sb-interstitial-induced defects generates a quasi-continuum of gap states in $Sb_2O_3$ nanocrystalline (Supplementary Notes 1, 2 and Supplementary Fig. 7-13). These states form a conductive pathway across the interface, allowing electrons to traverse the barrier via defect-mediated transport.

Cross-sectional TEM of the PVK/$C_{60}$/$Sb_2O_3$/indium zinc oxide (IZO) structure was performed to investigate the interlayer behavior of $Sb_2O_3$ (Fig. 1h and Supplementary Figs. 14, 15). A uniform 15 nm $Sb_2O_3$ film exhibits favorable interfacial contact with the amorphous $C_{60}$, owing to their low mutual chemical reactivity and residual stress (Supplementary Notes 3 and Supplementary Figs. 15–17). Notably, energy dispersive spectrometer (EDS) mapping confirms a well-defined layer structure and reveals negligible indium diffusion into the underlying layers, displaying the qualified sputtering tolerance of the $Sb_2O_3$ film (Fig. 1h and Supplementary Fig. 14). As shown in Supplementary Fig. 15, the nanocrystalline also exhibits a cubic structure, aligning with the planar TEM and XRD results (Fig. 1b, c). The structural stability also indicates its excellent sputtering tolerance and simple physic contact style with $C_{60}$ layer via van der Waals forces. Furthermore, we conducted dark photocurrent density-voltage ($J$–$V$) tests with device structure of ITO/Me-4PACz/PVK/$C_{60}$/with or without $Sb_2O_3$/IZO/Al (Supplementary Fig. 18). The $Sb_2O_3$ sample exhibits weaker leakage current, proving the excellent sputtering tolerance of $Sb_2O_3$[24].

### Buffer layer in PSCs

We applied $Sb_2O_3$ and $SnO_X$ as buffer layer in p-i-n PSCs (Fig. 2a), respectively, utilizing triple-cation $Cs_{0.05}(MA_{0.05}FA_{0.95})Pb(I_{0.88}Br_{0.12})_3$ perovskite with a bandgap of 1.64 eV fabricated via vacuum-solution hybrid method. Notably, the configuration represents the first implementation of $Sb_2O_3$ as a charge-transport layer in photovoltaics. We modulated the $C_{60}$ thickness via fixing the metal oxide thickness to 15 nm, as shown in Fig. 2b and Supplementary Fig. 19. The result of PV parameter distribution reveals a performance degradation with thinner $C_{60}$ in the $SnO_X$-based PSCs, marked by a sharp drop in fill factor (FF) at 5 nm. We attribute this primarily to the ligand exchange reaction between the TDMASn precursor and the perovskite (Supplementary Notes 4 and Supplementary Figs. 20–22). Besides, the 15 nm $C_{60}$ layer functions as an effective barrier, preventing the irreversible interfacial reaction in the $SnO_X$-based PSCs. In contrast, the $Sb_2O_3$-based PSCs present a broader process window for $C_{60}$ thickness, owing to the non-destructive nature of TE-$Sb_2O_3$.

The $Sb_2O_3$ thickness is further adjusted to 15 nm by using 5 nm $C_{60}$. We observed poor performance and repeatability for 5 nm $Sb_2O_3$ samples (Supplementary Fig. 23), which is ascribed to leakage current induced by non-denser 5 nm $Sb_2O_3$ (Supplementary Fig. 3). From reverse-scan $J$–$V$ curve (Fig. 2c), the champion PSC, using 5 nm $C_{60}$ and 15 nm $Sb_2O_3$, obtains a PCE of 23.18% with open-circuit voltage ($V_{OC}$) of 1.227 V, $J_{SC}$ of 22.34 mA/cm$^{-2}$ and FF of 84.55%, which has similar PV parameter with 23.17% of champion PSC with 15 nm $C_{60}$ and 15 nm $SnO_X$. Amazingly, the PCE marks the highest efficiency reported for mid/wide-bandgap PSCs using vacuum-solution hybrid method[6,25–39].

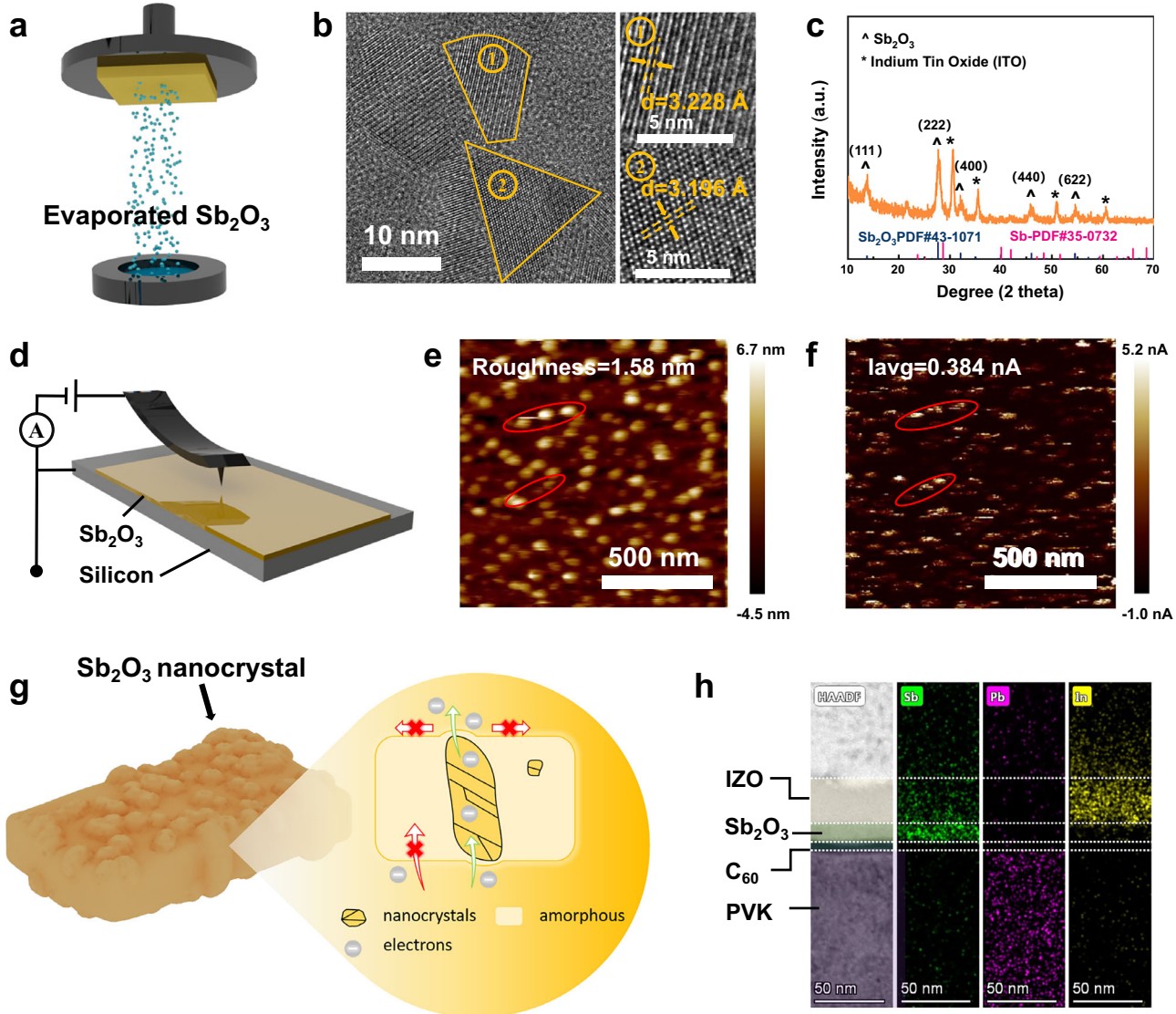

**Fig. 1 | Characterization of Sb₂O₃ films. a** Schematic illustration of the preparation of Sb₂O₃ by thermal evaporation method. **b** TEM image of the Sb₂O₃ film. **c** XRD spectrum of the Sb₂O₃ film. **d** Longitudinal c-AFM measurement setup. AFM (e) and corresponding c-AFM (f) images of the Sb₂O₃ film under a bias voltage of 3 V. g, Schematic illustration of conductive mechanism of the Sb₂O₃ film. h, Cross-sectional TEM image and the corresponding EDS mapping of PVK/C₆₀/Sb₂O₃/IZO stack films.

The stabilized power output (SPO) of the champion PSCs with $SnO_X$ and $Sb_2O_3$ also can be maintained at 22.74 and 23.02% after 600 s maximum power point (MPP) tracking, respectively (Fig. 2d). It illustrates that the $Sb_2O_3$ samples used by thin $C_{60}$ layer still have excellent photovoltaic performance. We assess its universality via fabricated PSCs with 1.59, 1.62, and 1.68 eV-bandgaps perovskite absorbers. The 1.59, 1.62, 1.68 eV-bandgaps PSCs deliver notable performance with champion PCEs of 22.28, 22.50, and 22.29%, respectively (Fig. 2e and Supplementary Table 2), which are also among the highest PCE for the hybrid deposited PSCs[6,25–39], as shown in Fig. 2f and Supplementary Table 3. All cells have achieved prominent FF, showing excellent generality in gently nature of $Sb_2O_3$. As shown in Supplementary Fig. 24, the integrated $J_{SC}$ from the external quantum efficiency (EQE) spectra of PSCs with 1.59, 1.62, 1.64, and 1.68 eV-bandgaps matched well with the $J_{SC}$ values from $J$–$V$ curves (Fig. 2c, e).

To gain a deep insight in the electrical transport mechanism of $Sb_2O_3$, the transient photocurrent (TPC) and transient photovoltage (TPV) measurements were performed in Fig. 2g, h. The comparable photoelectron lifetimes of $SnO_X$ and $Sb_2O_3$-based PSCs confirm the excellent efficiency of the defect-assisted carrier mechanism in $Sb_2O_3$.

The similar photovoltage lifetimes of them indicates that the defect-assisted transport enables efficient charge extraction without introducing additional non-radiative recombination.

**Optical benefit in PSTs**

To verify the adaptability of $Sb_2O_3$ in PSTs, we integrated the $Sb_2O_3$ film in 1 cm² PSTs, which manufactured double-textured silicon heterojunction bottom cells and p-i-n structured perovskite top cells with 1.64 eV of bandgap (Fig. 3a and Supplementary Fig. 25).

Based on the electrical feasibility in single-junction devices (Fig. 2b), we fabricated PSTs by varying the thickness of $C_{60}$ layers while fixing 15 nm $Sb_2O_3$ film. For direct comparison, control devices with 15 nm $C_{60}$ and 15 nm $SnO_X$ were prepared under identical conditions. The performance statistics of PSTs are summarized in Fig. 3b, c and Supplementary Fig. 26, wherein the gradually increased $J_{SC}$ with thinning $C_{60}$ for the tandem devices with $Sb_2O_3$, and more enhanced $J_{SC}$ than the tandem device with $SnO_X$ can be observed, facilitating to improving PCE. Among, the champion tandem device with $Sb_2O_3$ display a reverse-scan PCE of up to 30.28%, with a higher $J_{SC}$ of 20.26 mA/cm² than that of tandem device with $SnO_X$ (19.22 mA/cm²),

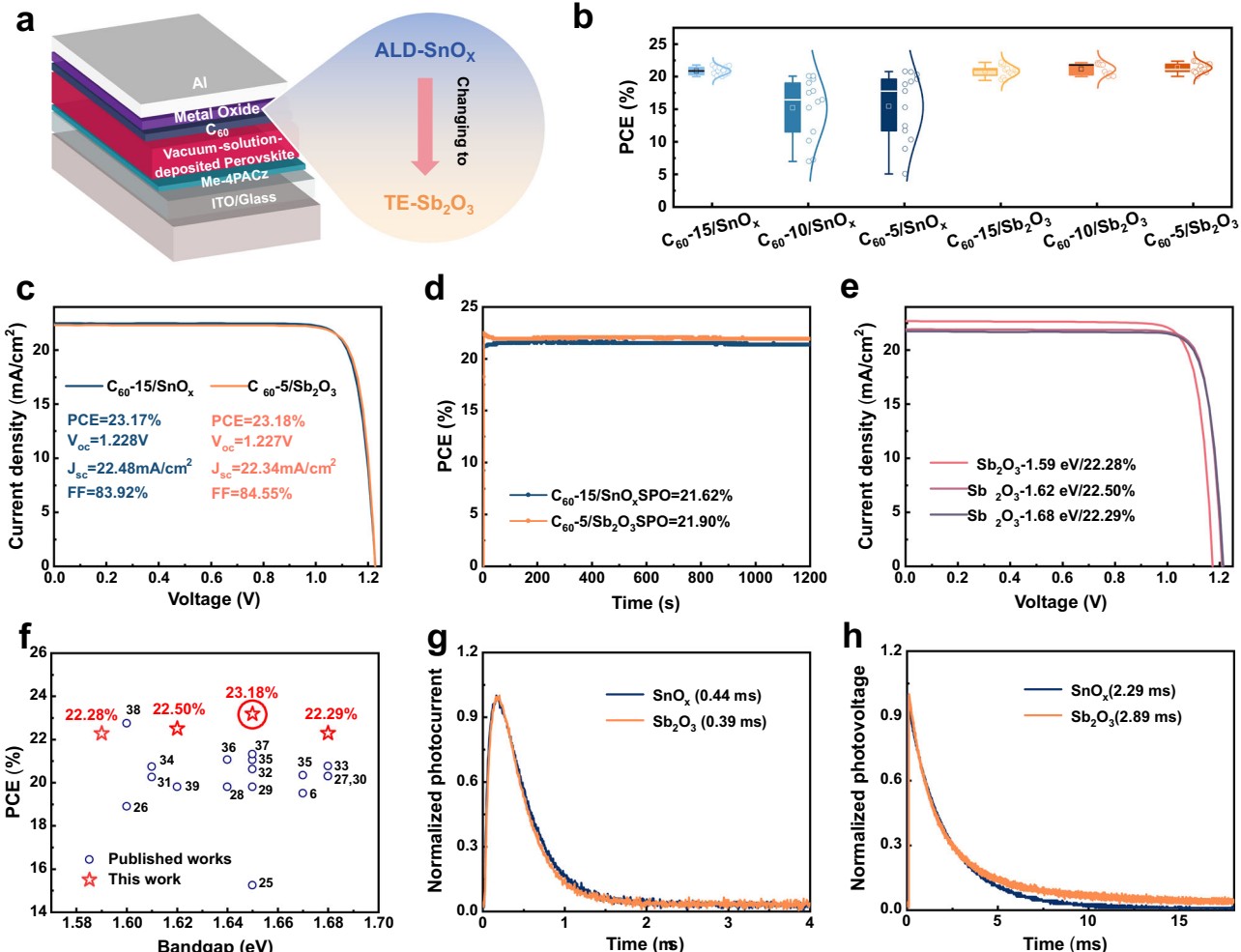

**Fig. 2 | Photovoltaic performance of PSCs based on Sb₂O₃ and SnOₓ.**
**a** Schematic structure of PSCs. **b** Statistics of PCE of the PSCs with different C₆₀ thicknesses based on SnOₓ and Sb₂O₃, respectively. For the box plots, the central line denotes the median, and the square denotes the mean. The box bounds represent the 25th and 75th percentiles. The solid lines extending above and below the box represent the maximum and minimum values, respectively. The reverse-scan *J*–*V* curves (**c**) and SPO (**d**) of the 1.64 eV-bandgap champion PSCs based on SnOₓ and Sb₂O₃ under AM1.5 G irradiation. e, The reverse-scan *J*–*V* curves of the 1.59, 1.62, 1.68 eV-bandgaps champion PSCs based on Sb₂O₃ under AM1.5 G irradiation. **f** Evolution of published PCE of mid/wide-bandgap PSCs prepared via vacuum-solution hybrid deposition[6,25–39]. TPC (**g**) and TPV (**h**) measurements of SnOₓ- and Sb₂O₃-based PSCs.

and almost the same $V_{OC}$ of 1.894 V and FF of 78.67% (Fig. 3d and Supplementary Table 4). The SPO of 30.1% was also recorded after 2400 s MPP tracking (Fig. 3e).

Further, we conducted EQE and ultraviolet-visible (UV-vis) measurements to explain the reason of improving $J_{SC}$ and color difference in aperture area (Fig. 3f, g and Supplementary Figs. 27, 28). The UV-vis shows reflection, transmittance and absorbance spectra of different thicknesses of C₆₀. We detected that the average transmission ($T_{avg}$) decreases from 81.17 to 63.87% in the wavelength range from 300 to 560 nm, with increasing C₆₀ film thickness from 5 to 15 nm. As expected, the intensity of absorption increases with increasing film thickness. Therefore, the stronger EQE response of top sub-cell at the 300-560 nm wavelength is exhibited with thinning C₆₀ film, resulting in a higher photocurrent of 20.45 mA/cm² for C₆₀ (5 nm)/Sb₂O₃ tandem device of top sub-cell and well-matched photocurrent of 20.54 mA/cm² for bottom sub-cell (Fig. 3f). Next, UV-vis was employed to investigate the optical properties of C₆₀/SnOₓ and C₆₀/Sb₂O₃ stacks (Fig. 3g and Supplementary Fig. 28). The C₆₀ (5 nm)/Sb₂O₃ (15 nm) sample shows an $T_{avg}$ of 74.85%, exceeding that of PVK/C₆₀ (15 nm)/ SnOₓ (15 nm) sample (62.68%) in the wavelength range from 300 to 560 nm. Similarly, the intensity of absorption for C₆₀ (5 nm)/Sb₂O₃ (15 nm) sample is lower than C₆₀ (15 nm)/SnOₓ (15 nm) sample,

suggesting the achievement of lower parasitic light absorption by thinning C₆₀.

To evaluate devices stability under operational conditions (Supplementary Notes 5 and Supplementary Fig. 29-39), we performed MPP tracking on encapsulated Sb₂O₃-based PST (Fig. 3h). After 500 h under white light emitting diode (LED) illumination (100 mW/cm²), The Sb₂O₃-based device exhibited negligible efficiency degradation. Furthermore, both encapsulated devices conducted the accelerated aging tests (LED illumination or 65 °C over 1000 h), retained over 90% of their initial PCE (Supplementary Figs. 37, 38).

**Application in encapsulated large-area PSTs**

Scaling up aperture area in tandem devices represent a necessary trend for commercial development. To examine the homogeneity of Sb₂O₃, we deposited a 45 nm Sb₂O₃ film on a 10 × 10 cm² glass substrate, which was subsequently divide it into 25 sub-samples, each measuring 2 × 2 cm² (Fig. 4a). The results indicate nearly identical thickness across all samples, revealing superior uniformity of the large-area Sb₂O₃ film deposition in a 100 cm² region (Fig. 4b and Supplementary Fig. 40). Besides, we further selected five representative positions to verify the uniformity of C₆₀ and the thinner Sb₂O₃ (15 nm) layers (Supplementary Notes 6 and Supplementary Figs. 41–43,

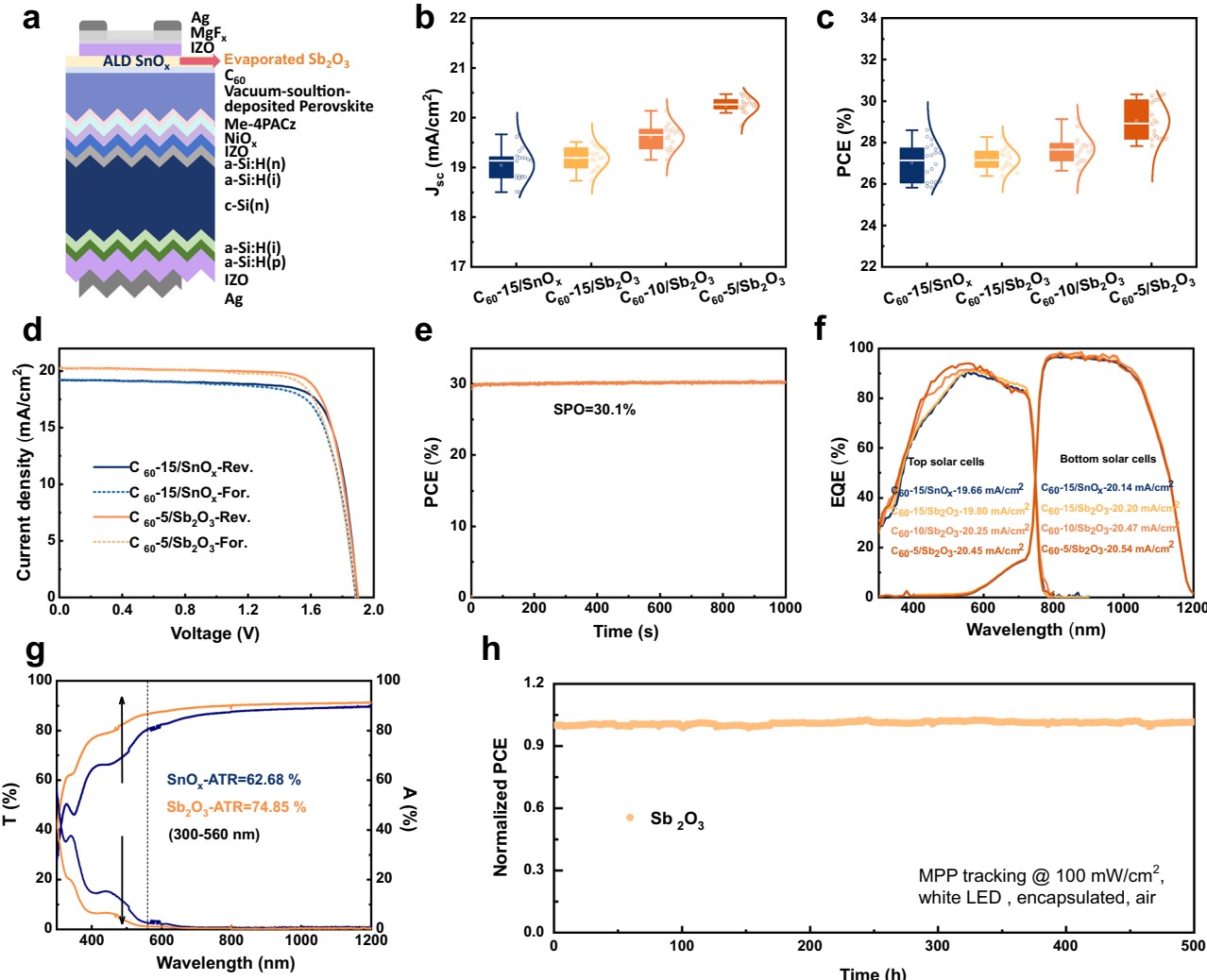

**Fig. 3 | Photovoltaic performance and stability of PSTs based on Sb₂O₃ and SnOₓ.** **a** Schematic of PST structure based on a double-side and sub-micron-pyramid-structured silicon heterojunction cell. Statistics of PCE (**b**) and $J_{SC}$ (**c**) of the PSTs with Sb₂O₃ and SnOₓ. For the box plots, the central line denotes the median, and the square denotes the mean. The box bounds represent the 25th and 75th percentiles. The solid lines extending above and below the box represent the maximum and minimum values, respectively. $J$–$V$ curves (**d**) and SPO (**e**) of the 1 cm² champion PSTs. **f** EQE spectra of PSTs with Sb₂O₃ and SnOₓ. **g** Absorptance and transmittance spectra of C₆₀/Sb₂O₃ and C₆₀/SnOₓ films. **h** Normalized *PCE* of encapsulated Sb₂O₃-based PST for MPP tracking under a white LED lamp illumination at 100 mW/cm² in air.

Supplementary Table 5). The normalized PCE distribution of 100 sub-cells on a 10 × 10 cm² substrate was also shown in Fig. 4c. Similarly, we fabricated Sb₂O₃ films as buffer layer in 100 sub-cells, adopting the device structure illustrated in Fig. 2a. The high similarity of the PCE for all sub-cells further proves the homogeneity of Sb₂O₃ film.

Furthermore, we used vacuum-solution hybrid method to fabricate 110.25 cm² encapsulated PSTs with an aperture of 64.64 cm² and achieved an enhanced PCE from 27.29 to 28.16% for Sb₂O₃-based PST, which is attributed to an improvement of $J_{SC}$ from 18.61 to 18.96 mA/cm² (Fig. 4d, Supplementary Fig. 44 and Supplementary Table 6). Crucially, The PCE of 28.16% is currently among the highest PCE for larger-area PSTs with the aperture area over 10 cm² in published literature[33,39–49] (Fig. 4e and Supplementary Table 7). Moreover, the encapsulated large-area PST was further certified by a third party (National Institute of Metrology) showed a reverse-scan PCE of 27.70% (Supplementary Fig. 45). Furthermore, a SPO of 27.66% was confirmed following 1800 s of stabilization at the MPP (Supplementary Fig. 46). In addition, the process demonstrates excellent reproducibility across multiple independent fabrication, indicating promising scalability (Supplementary Notes 7, Supplementary Fig. 47 and Supplementary Table 8). Notably,

the development of Sb₂O₃ achieves dual breakthroughs in performance and cost, which reduces fabrication cost to ~70% of conventional ALD-SnOₓ, demonstrating significant commercial potential (Supplementary Notes 8 and Supplementary Table 9).

## Disscussion

In this work, we firstly developed an amorphous-nanocrystalline Sb₂O₃ film via thermal evaporation to serve as an alternative buffer layer to ALD-deposited SnOₓ, where the embedded nanocrystals form effective electron transport pathways. The Sb₂O₃ film exhibits exceptional optical transmission, high sputtering tolerance, and excellent compatibility with ultra-thin C₆₀ (5 nm), enabling non-destructive perovskite integration. Using a vacuum-solution hybrid processing approach, Sb₂O₃-based PSCs with tunable bandgaps (1.59, 1.62, 1.64, and 1.68 eV) achieved PCEs of 22.28, 22.50, 23.18, and 22.29%, respectively. Notably, the 23.18% PCE currently represents the highest reported value for mid/wide-bandgap PSCs fabricated using the vacuum-solution hybrid method. Furthermore, Sb₂O₃-based PSTs have achieved a champion PCE of 30.28% for 1.0 cm² devices, surpassing the 28.69% efficiency demonstrated by SnOₓ-based PSTs. The 64.64 cm² encapsulated PSTs with Sb₂O₃ buffer

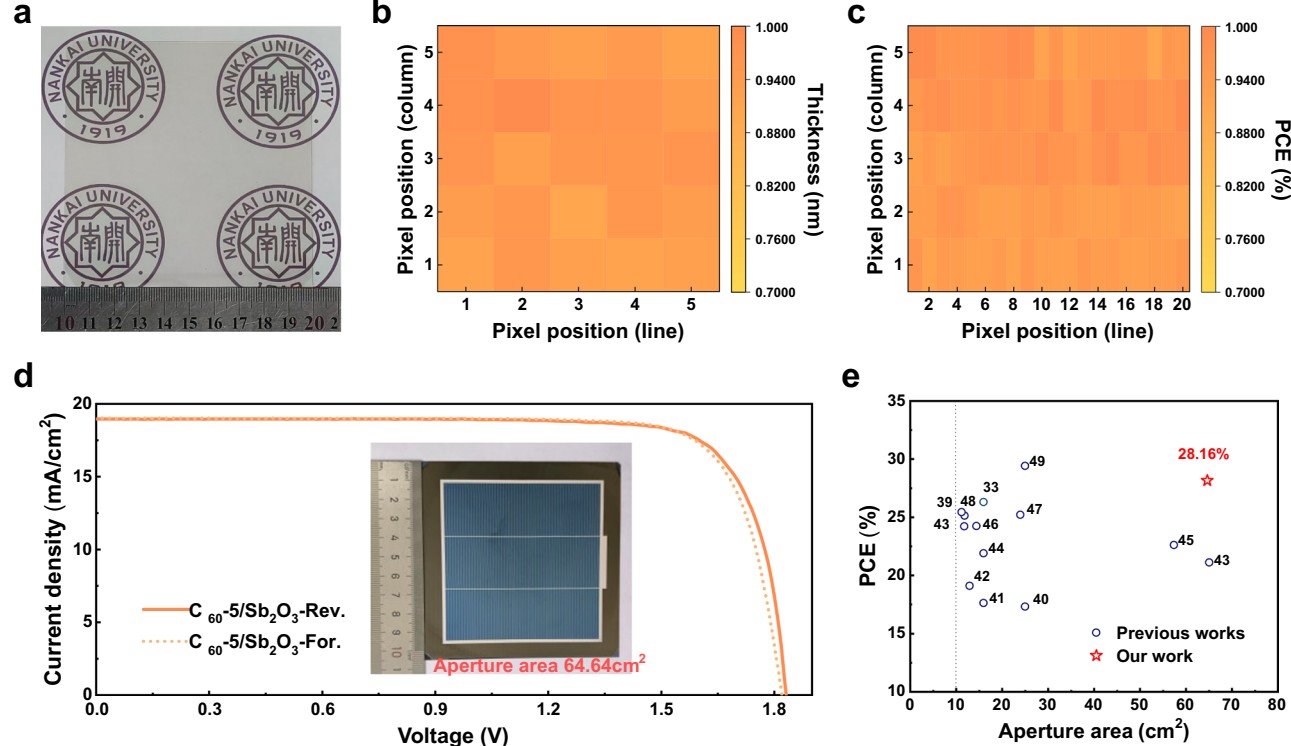

**Fig. 4 | Photovoltaic performance of large-area tandem solar cells based on Sb$_2$O$_3$ and SnO$_X$. a** Photograph of Sb$_2$O$_3$ on a 10 × 10 cm$^2$ glass substrate. **b** Statistical distribution of normalized thicknesses with 25 sub-samples on a 10 × 10 cm$^2$ glass substrate. **c**, Statistical distribution of normalized PCE with 100 sub-cells based on Sb$_2$O$_3$ on a 10 × 10 cm$^2$ substrate. **d** The $J$–$V$ curves of the encapsulated large-area champion PST (64.64 cm$^2$) with Sb$_2$O$_3$. **e**, Evolution of published PCE of PSTs (>10 cm$^2$)[33,39–49].

layer demonstrate outstanding performance, achieving a champion PCE of 28.16% (with a certified PCE of 27.70%). This represents one of the highest reported efficiencies for larger-area PSTs with an aperture area exceeding 10 cm$^2$ in published literature. Enabled by Sb$_2$O$_3$ films, the optical parasitic loss from C$_{60}$ in PSTs is further reduced, paving the way for efficiencies beyond 35% in the future.

## Methods

### Materials

Me-4PACz (>99%) and C$_{60}$ were purchased from Lumtec. Ethanol (99.5% anhydrous) and isopropyl alcohol (IPA) (99.5% anhydrous) were purchased from Alfa Aesar. Lead iodide (PbI$_2$, >99.99%) and antimony trioxide (Sb$_2$O$_3$, >99.9%) were purchased from Sigma-Aldrich. Formamidine bromide (FABr), formamidine iodide (FAI), methylamine hydrochloride (MACl), Cesium chloride (CsCl), 4-fluorobenzylamine hydroiodide (F-PMAI) and 1,3-propanediammonium iodide (PDADI) were purchased from Xi'an Polymer Light Technology (China).

### Fabrication of silicon bottom cells

We employed a fabrication process for the silicon heterojunction (SHJ) bottom cell that closely followed established methods reported in the literature[5]. The procedure is as follows: an n-type <100> CZ Si wafer with a resistivity of 1–5 Ω·cm was used. An initial etching step was conducted to remove saw damage and polish the wafer. Subsequently, both the front and rear surfaces of the wafer were mildly textured using a relatively low-concentration alkaline solution to ensure uniform small-pyramid structures with a size of 300–500 nm on both sides. After standard cleaning and a hydrogen fluoride dip, hydrogenated intrinsic amorphous silicon layers (~5 nm thick) were deposited on both sides. Then, an n-type hydrogenated nanocrystalline silicon oxide layer (~15 nm) and a p-type hydrogenated nanocrystalline silicon (nc-Si:H) layer (~20 nm) were deposited on the front

and rear, respectively. To complete the bottom cell, an 80-nm-thick In$_2$O$_3$-based transparent conductive oxide followed by silver was deposited on the rear, while a 10-nm-thick TCO film was deposited on the front. The electrode area was 1.1 × 1.1 cm$^2$, and the silicon cells were laser-cut to 2.03 × 2.03 cm$^2$ substrates for tandem fabrication.

### Fabrication of single-junction PSCs

The single-junction devices were fabricated with the structure: ITO/Me-4PACz/Perovskite/PDADI/C$_{60}$/SnO$_X$ or Sb$_2$O$_3$/Al. Glass substrates with ITO coating were cleaned with three solvents in the sequence of deionized water, acetone, and ethanol. The Me-4PACz solution in ethanol solution was spin-coated at 3000 rpm for 30 s, following an annealing treatment at 100 °C for 10 min in N$_2$ glovebox. Then, the Me-4PACz film was clean up via spin-coating 100 μL ethanol at 5000 rpm for 30 s. CsCl was evaporated in a Lesker mini Spectros system chamber (working pressure < 5 × 10$^{-6}$ mbar, evaporation rates of 0.4 Å/s). Next, PbI$_2$ was co-evaporated with CsCl to form inorganic precursor in same chamber (working pressure < 5 × 10$^{-6}$ mbar, evaporation rates of 2 Å/s for PbI$_2$ and of 0.1 Å/s for CsCl). The substrate temperature was kept at 30 °C. Subsequently, a solution of FABr and FAI with different molar ratio of 1:5.2, 4.6, 1.5, 0.8 (bandgaps of 1.59, 1.62, 1.64, 1.68 eV) dissolved in ethanol and containing 10 mol% of MACl and 5 mol% of F-PMAI, was spin-coated at 5000 rpm for 30 s in N$_2$ glovebox, following by the annealing step at 150 °C for 20 min in ambient air with a 30-35% humidity. PDADI dissolved in IPA (1 mg mL$^{-1}$) was spin-coated onto perovskite layer at 5000 rpm for 30 s, following an annealing treatment at 100 °C for 5 min in N$_2$ glovebox. Afterward, different thicknesses of C$_{60}$ (5, 10, 15 nm) and of Sb$_2$O$_3$ (5, 15, 25 nm) were sequentially evaporated in a homemade evaporation system to fabricate Sb$_2$O$_3$-based PSCs (working pressure < 5 × 10$^{-6}$ mbar, substrate holder at 30 °C). For the PSCs with SnO$_x$, the Sb$_2$O$_3$ is replaced by 15 nm SnO$_X$ through ALD at 80 °C using TDMASn and hydrogen peroxide as precursors. Finally, the metal grids

were fabricated by depositing a 80 nm thick layer of Al (Angstrom Engineering) through thermal evaporation, using a shadow mask. Besides, the active area was 0.0755 cm² for single-junction devices.

## Fabrication of PSTs

Twenty nm nickel oxide ($NiO_X$) layer was sputtered using a nickle target at room temperature in pure Ar at 80 W to cap the ITO recombination layer on the top of SHJ bottom cells. Then, Me-4PACz, perovskite, PDADI, $C_{60}$ and $Sb_2O_3$ or are deposited on $NiO_X$ substrate using same process as the single-junction PSCs, where the perovskite with 1.64 eV bandgap is used in small-area tandem devices (1 cm²), the perovskite with 1.59 eV bandgap is used in large-area tandem devices (64.64 cm²). Subsequently, 50 nm of IZO transparent electrode was sputtered in a Lesker sputtering system using a 6-in. target (90% $In_2O_3$ and 10% ZnO) with a radio frequency power of 38 W (sheet resistance of 40 Ω/sq). Afterward, 200 nm of Ag was evaporated through a shadow mask to form the metal electrode for small-area tandem devices (1 cm²), and the screen-printing metal grids instead of the evaporated Ag electrode for large-area tandem devices (64.64 cm²), following the curing of silver paste was annealing at 85 °C for 20 min.

## Device characterization

$J$–$V$ curves of single-junction PSCs and tandem solar cells were measured by a Keithley 2400 Source meter under Enli Solar Simulator (AM 1.5 G, 100 mW cm⁻²), calibrated by a standard silicon reference cell. The devices were measured with a scan rate of 20 mV/s, the voltage step of 0.02 V, and delay time of 40 ms. During the J–V test, shadow masks with area of 0.0755 cm² for single-junction and of 1 or 64.64 cm² for tandem solar cells were used, respectively. EQE of single-junction PSCs were measured using an Enli Tec (Taiwan) system. For tandem solar cells, EQE spectra were measured by a QEX10 PV Measurement system in the wavelength from 300 to 1200 nm with a scanning rate of 10 nm. SPO was measured in $N_2$ glovebox at room temperature with multichannel system (Enli. Tech.) under AM 1.5 G illumination by a xenonlamp. Device stability was evaluated through aging tests under LED illumination using a high-power LED light source system (CEL-LED100HA-96, AuLight) and full-spectrum AM 1.5 G illumination provided by a xenon-lamp solar simulator (FG-MN001-ACA-AM1.5G-100, FengGuan), each calibrated to 100 mW/cm². Encapsulated tandem devices were subjected to continuous light soaking at ~40 °C under both illumination conditions, with periodic J–V measurements to track the evolution of MPP efficiency over time.

## Other characterization

XPS and UPS were performed by Thermo Scientific ESCALAB 250Xi spectrometer. UV-vis spectra was carried out through a Varian Cary 500 spectrophotometer. SEM images was characterized by FEINano-SEM650 with an acceleration voltage of 1 kV. AFM, kelvin probe force microscopy (KPFM) and c-AFM tested by Bruker Dimension Icon with Pt/Ir coated conductive probes. Grazing-incidence X-ray diffraction (GIXRD) patterns of films were obtained by a multifunctional diffractometer (Rigaku, ATX-XRD) with Cu Kα radiation ($\lambda$ = 1.5405 Å). The cross-sectional TEM samples were prepared using a focused ion beam (FIB) lift-out technique on a ZEISS Crossbeam 540 FIB-SEM system. TEM imaging and selected area electron diffraction analysis were performed using an FEI Tecnai F20 field-emission transmission electron microscope operating at an acceleration voltage of 200 kV. Dark $J$–$V$ curve measurement was conducted in dark in dark by Keithley 2400 Source meter at a scan rate of 20 mV/s with a delay time of 20 ms, where the bias scan ranged from −2 to 2 V. TPC and TPV were obtained with a digital oscilloscope (DOS-X 304 A).

## Reporting summary

Further information on research design is available in the Nature Portfolio Reporting Summary linked to this article.

## Data availability

Source data are provided with this paper. All other data of this work are available from the corresponding authors on request. Source data are provided with this paper.

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

## Acknowledgements

The authors are gratefully acknowledged for the financial support of the National Key Research and Development Program of China (Grant No. 2023YFB4202503), the Joint Funds of the National Natural Science Foundation of China (Grant No. U21A2072), the National Natural Science Foundation of China (Grant No. 62274099), the Overseas Expertise Introduction Project for Discipline Innovation of Higher Education of China (Grant No. B16027), Yunnan Provincial Science and Technology Project at Southwest United Graduate School (Grant No. 202302A0370009), Tianjin Science and Technology Project (Grant No. 24ZXZSSS00160), the Haihe Laboratory of Sustainable Chemical Transformations and the Fundamental Research Funds for the Central Universities, Nankai University.

## Author contributions

B.S. and Z.S. contributed equally to this work and proposed the research and designed the experiments. P.L. help with the preparation of screen-printing metal grids. W.H. and R.K. help with the fabrication of transparent electrodes. Yi.L. and C.S. help with the fabrication of the ALD buffer layer. Yu.L. helped the fabrication of nickel oxide hole transfer layer. D.Z. helped the fabrication of metal electrode.Xi.Z., B.S., X.W., Z.Z., and X.D. provided testing help. 1 $cm^2$ bottom cell development and fabrication by F.Z., M.Y, Y.H., B.H., and X.X. 64.64 $cm^2$ bottom cell development and fabrication by R.X, Xu.Z., Y.C., and J.G. Z.S. wrote the first version of the manuscript. Xi.Z., F.X. and S.D.W. revised the manuscript. Xi.Z. directed the overall project. Correspondence and requests for materials should be addressed to Xi.Z.

## Competing interests

The authors declare no competing interests.
