## [Transparent Peer Review file · Nature Communications]

Antimony Oxide Buffer Layer for Single- and Double-Junction Perovskite-Based Solar Cells

Corresponding Author: Professor Xiaodan Zhang

Version 0:

Reviewer comments:

Reviewer #1

(Remarks to the Author)

This manuscript investigates the application of antimony trioxide (Sb_2O_3) thin films as buffer layers in perovskite/silicon tandem solar cells, comparing them with conventional atomic layer deposition-based tin oxide (SnO_x). The findings indicate that Sb_2O_3 thin films exhibit excellent optical transmittance, high sputtering resistance, and compatibility with ultrathin C60, effectively reducing optical parasitic losses and enhancing device performance. Overall, the concepts and results presented herein are promising. Reducing the optical parasitic absorption of functional layers has been a significant research direction in this field; however, several issues require further consideration:

1. Given the proximity of the constituent oxides in the periodic table, what is the underlying mechanism of perovskite film chemical etching by SnO_x ? Moreover, what mechanism allows Sb_2O_3 to resist such chemical etching?
2. Does the thermal evaporation of Sb_2O_3 result in the formation of metallic antimony? The presence of metallic antimony could potentially contribute to the n-type characteristics of Sb_2O_3 .
3. Based on the current experimental data, the amorphous-nanocrystalline vertical conduction mode in Sb_2O_3 is not entirely convincing. Could the authors provide additional supporting evidence?
4. Reducing the fullerene thickness is crucial for minimizing optical parasitic absorption. However, what is the coverage of a 5 nm C60 layer? Does this affect electron extraction efficiency?
5. Figure S8 contains a clear error; please review it thoroughly. Additionally, the valence band edge truncation in Figure S6 appears questionable. It is recommended to increase the number of experimental samples to ensure statistically robust results.
6. Since the primary advantage of Sb_2O_3 over SnO_x in this study is the reduced chemical etching of perovskite films, the results of various stability tests are critical. Stability data under continuous illumination at maximum power point tracking, temperature, and electrical bias should be included for devices incorporating Sb_2O_3 buffer layers, along with appropriate explanations.

Reviewer #2

(Remarks to the Author)

This work reports a promising strategy of replacing ALD-deposited SnO_x layer with an Sb_2O_3 layer to reduce optical losses in the C60/ SnO_x stacked layer. This approach reduces the thickness of the C60 layer, decreases the parasitic absorption and finally increases the J_{sc} , and protects the perovskite layer. This strategy is universal to a broad range of perovskite solar cells with various bandgaps and to large-area tandem solar cells. This work is of great interest and significance to the community of perovskite single-junction and tandem solar cells, especially this work provides an alternative thermal evaporation deposition technique to the existing ALD technique, which could reduce the fabrication cost. Therefore, I would recommend publication of this manuscript after addressing the following minor issues.

- (1) As well known, C60 could increase the fabrication cost of perovskite solar cells. Completely substituting C60 seems a great challenge for perovskite solar cells. Could the authors comment on whether Sb_2O_3 can replace C60 and work as an ETL in tandem cells?
- (2) The authors claim that the higher mobility of Sb_2O_3 is attributed to more favorable energy level alignment. Please explain more on this.
- (3) In Figure 2d and 2f, please clarify the voltage scan direction. In Figure 2h, please clarify what the current densities correspond to.
- (4) Section 5 of Page 10: to examine the homogeneity of Sb_2O_3 , the authors deposited a 45 nm Sb_2O_3 layer on $10 \times 10 \text{ cm}^2$

glass substrate. Could the authors comment on uniformity of the 15 nm thick Sb₂O₃ for scaling up cells?

(5) Page 8: to evaluate the optical benefits of Sb₂O₃ layer and reduced C60 thickness, the authors should compare the average transmittance of C60/Sb₂O₃ and C60/SnO_x stacked layers.

Reviewer #3

(Remarks to the Author)

Shi et al. introduce an antimony oxide as buffer layer for perovskite based single junction and tandem solar cells. This approach enhances the power conversion efficiency (PCE) and stability of perovskite/silicon tandem cells. The optimized tandem device achieves a PCE of 30.1%. These results underscore the strategy's potential for industrial-scale applications. However, the stability and efficiency of the devices in the article is not that impressive to guarantee its publication in Nature Communications. Overall, this work demonstrates interesting result with good material design. Therefore, I recommend this paper to be reconsidered after revisions and suggest some points for further improvement.

1. The maximum power point testing should be incorporated in the manuscript with certain duration. Additionally, post-aging microstructural analysis using SEM, XRD and TOF-SIMS should be applied correspondingly.
2. The TEM image in Fig. 1g appears blurred, and the interfaces are difficult to distinguish. A low-magnification TEM image should be provided to clearly demonstrate the long-range behavior of the interlayer.
3. The AFM results show that the RMS roughness decreases after Sb₂O₃ deposition compared to the reference film. Does this imply that Sb₂O₃ does not form a uniform coverage?
4. The explanation of the electron transfer mechanism remains unclear. Additional characterizations are necessary to elucidate the underlying charge transfer behavior in the Sb₂O₃ layer.
5. The nature of the Sb₂O₃/C60 interface is not well defined. It should be clarified whether the interface is governed by simple physical contact or involves specific chemical interactions.
6. It should be specified whether the XPS and UPS measurements were conducted on the Sb₂O₃ film alone or on the C60/Sb₂O₃ stack, considering the limited detection depth of these techniques. Any potential band bending effects at the interface should also be discussed.
7. Although the IZO layer thickness is claimed to exceed 50 nm, it is not clearly visible in the SEM image of the perovskite/IZO film. TOF-SIMS measurements may provide more insightful information to correlate potential IZO damage with changes in the perovskite morphology.
8. The reference device employing a SnO₂ buffer layer shows a PCE of only ~28%, which is noticeably lower than previously reported values from the same research group. The reason for this performance decline should be clarified.
9. The intrinsic stability of the Sb₂O₃ layer has not been discussed. Additionally, its potential impact on the long-term stability (under thermal treatment, light illumination, and moisture) of the underlying perovskite film remains unclear and should be addressed.
10. Related to 9, the device stability under various stresses should also be addressed. At least, for the single junction devices.
11. The mechanical toughness of the developed interface remains unclear. Additional analysis is needed to evaluate its resistance to delamination or mechanical stress, particularly in the context of scalable device fabrication and long-term operation.
12. The manuscript lacks a detailed comparison with other potential buffer layer materials that are frequently used. A discussion on how Sb₂O₃ performs relative to these commonly used alternatives would strengthen the manuscript.
13. A thorough evaluation, supported by quantitative data, is necessary to assess the economic and technical feasibility of implementing the Sb₂O₃ interlayer in large-scale manufacturing, especially in comparison with commonly adopted strategies.
14. For the certification of the large-area tandem device, data on hysteresis behavior and steady-state power output are missing. These measurements are essential to validate the reliability and performance consistency of the device.
15. The device performance results using different bandgap perovskites primarily serve to demonstrate generality and would be more appropriate in the Supplementary Information. In contrast, the comparison of electronic properties should be presented in the main manuscript to support the core scientific discussion.
16. The light stability results show no significant difference between the SnO₂- and Sb₂O₃-based tandem devices. If SnO₂ were causing substantial damage to the perovskite surface, a corresponding impact on device stability would be expected. This apparent inconsistency should be addressed and clarified.

Version 1:

Reviewer comments:

Reviewer #1

(Remarks to the Author)

The revised manuscript has addressed all the reviewers' concerns, and I'm pleased to accept it for publication.

Reviewer #2

(Remarks to the Author)

The authors have addressed the comments and issues raised by the reviewers. I am satisfied with their response and revision. I would recommend acceptance of this manuscript at its current version.

Reviewer #3

(Remarks to the Author)

The manuscript has been improved significantly. However, several important issues remain insufficiently addressed, particularly regarding the tandem characterization, e.g., the textured Si bottom cell, and the functional layers in the full stack.

1. In this work, long-term stability and MPP tracking are performed under a 100 mW cm⁻² white LED source, for single-junction and especially for tandem devices. While this is experimentally convenient, it raises serious concerns for monolithic perovskite/Si tandems, which is not acceptable.

Simply, the spectral distribution of a "white LED" typically differs from AM 1.5G, particularly in the blue and near-IR regions that are critical for current matching between the perovskite top and Si bottom cells, and bias subcells to affect its efficiency decay dynamics. I therefore recommend that the authors to Demonstrate stability under AM 1.5G. Without such analysis, the reported "negligible efficiency loss" under LED illumination cannot be directly interpreted as stability under realistic PV operating conditions.

2. The tandem devices are based on double-textured SHJ bottom cells with "sub-micron-pyramid-structured" surfaces. However, the manuscript does not provide quantitative information on this texture, e.g., pyramid height, base width, pitch, or RMS roughness. Moreover, the etching process (e.g., alkaline vs acidic texturing, etch time, orientation dependence) is not described in enough detail to allow reproducibility.

3. Most of the detailed structural and electronic characterization of Sb₂O₃ (AFM, TEM, GIXRD, c-AFM, UPS, defect-state analysis) is performed on planar substrates or planar stacks. In contrast, the actual tandem devices use Sb₂O₃ deposited on a NiO_x/ITO-coated, pyramid-textured SHJ surface, followed by sputtered IZO and Ag or screen-printed contacts. I suggest the authors providing detailed characterizations on the actual textured tandem stack along and across pyramids, showing continuity and thickness of Sb₂O₃ over the entire facet and its crystallinity (nanocrystalline as claimed in the manuscript).

4. Regarding Large-area uniformity, the PCE distribution of sufficient numbers of small cells on a 10×10 cm² substrate, are used to support scalability. Please also comment on the yield and variability among large-area devices fabricated under nominally identical conditions.

5. The authors attribute the inferior long-term stability of Sb₂O₃-based PSTs to poorer compactness and facilitated ion migration through the molecular-crystal Sb₂O₃. Please provide direct evidence to support the claim.

6. Please ensure that all acronyms (TE, PST, PVK, etc.) are defined at first use.

Version 2:

Reviewer comments:

Reviewer #3

(Remarks to the Author)

I appreciate the authors' efforts in addressing the majority of the concerns raised in the previous round, particularly regarding the large-area yield statistics and the structural characterization of the silicon bottom cell.

However, the refusal to validate device stability under standard AM 1.5G conditions remains a critical methodological barrier that prevents publication. For monolithic perovskite/silicon tandems, the spectral mismatch of white LEDs fundamentally alters the current-matching condition, forcing sub-cells into artificial bias states that differ significantly from real-world operation. This renders your degradation analysis unreliable, as the perovskite top-cell may not be under the limiting stress encountered in the field. Labeling the work as "pioneering" does not exempt it from basic metrological standards. Without AM 1.5G MPP tracking, I think the claim of device stability is unsubstantiated and the manuscript remains unsuitable for publication.

Version 3:

Reviewer comments:

Reviewer #1

(Remarks to the Author)

I have satisfied with all the responses.

Reviewer #3

(Remarks to the Author)

I appreciate the authors' efforts. I'm satisfied with this revision. The manuscript is ready for publication.

Reviewer #4

(Remarks to the Author)

The authors have addressed the comments and issues raised by the reviewers. I am satisfied with their response and revision. I would recommend acceptance of this manuscript at its current version.

Point-by-point response to the reviewer comments

Reviewer #1 (Remarks to the Author):

This manuscript investigates the application of antimony trioxide (Sb_2O_3) thin films as buffer layers in perovskite/silicon tandem solar cells, comparing them with conventional atomic layer deposition-based tin oxide (SnO_x). The findings indicate that Sb_2O_3 thin films exhibit excellent optical transmittance, high sputtering resistance, and compatibility with ultrathin C_{60} , effectively reducing optical parasitic losses and enhancing device performance. Overall, the concepts and results presented herein are promising. Reducing the optical parasitic absorption of functional layers has been a significant research direction in this field; however, several issues require further consideration:

Comment #1. Given the proximity of the constituent oxides in the periodic table, what is the underlying mechanism of perovskite film chemical etching by SnO_x ? Moreover, what mechanism allows Sb_2O_3 to resist such chemical etching?

Response: Thank you for the valuable comments. ALD- SnO_x is typically fabricated using TDMA_{Sn} and H_2O as precursors. The whole ALD process involves a high-temperature and water-rich environment, which is extremely unfavorable to perovskite. Critically, the TDMA_{Sn} precursor undergoes a severe **ligand exchange reaction** with the perovskite, driven by the displacement of its dimethylamine groups by formamidinium ions¹. As evidenced by the N 1s XPS spectrum (**Fig. R1**), the emergence of a distinct peak at ~ 398.5 eV alongside the characteristic perovskite signal at ~ 400.6 eV confirms the formation of reduced nitrogen species via Sn–N bonding^{2,3}. In addition, the emergence of PbI_2 characteristic peak confirms this result in the XRD pattern (**Fig. R2**). In contrast, thermal evaporated Sb_2O_3 is a purely physical process involving sublimation and condensation⁴,

without complex precursors or post-deposition annealing. This inherent simplicity prevents any chemical interaction with the perovskite layer, as confirmed by XPS and XRD results (Fig. R1-2).

Fig. R1 XPS spectra of N 1s, I 3d, Pb 4f, Sb 3d, Sn 3d core levels for the pristine PVK and PVK/SnO_x or Sb₂O₃.

Fig. R2 XRD patterns of pristine PVK and PVK/SnO_x or Sb₂O₃. at different incidence angles (θ_{inc} =0.3, 0.4 and 1°).

References

- 1 Palmstrom, A. F. *et al.* Interfacial effects of tin oxide atomic layer deposition in metal halide perovskite photovoltaics. *Advanced Energy Materials* **8**, 1800591 (2018).
<https://doi.org/10.1002/aenm.201800591>
- 2 Zhou, B., Zhou, W. & Wu, P. Ferromagnetic ordering and metallic-like conductivity in sputtered SnNx films. *Journal of Alloys and Compounds* **604**, 106-111 (2014).
<https://doi.org/10.1016/j.jallcom.2014.03.098>
- 3 Hultqvist, A. *et al.* SnO_x atomic layer deposition on bare perovskite—an investigation of initial growth dynamics, interface chemistry, and solar cell performance. *ACS Applied Energy Materials* **4**, 510-522 (2021). <https://doi.org/10.1021/acsaem.0c02405>
- 4 Liu, K. *et al.* A wafer-scale van der Waals dielectric made from an inorganic molecular crystal film. *Nature Electronics* **4**, 906-913 (2021). <https://doi.org/10.1038/s41928-021-00683-w>

Comment #2. Does the thermal evaporation of Sb_2O_3 result in the formation of metallic antimony? The presence of metallic antimony could potentially contribute to the n-type characteristics of Sb_2O_3 .

Response: Thank you for the valuable comments regarding the potential formation of metallic antimony (Sb^0) and its contribution in the n-type character for Sb_2O_3 film. Our XPS data confirm the presence of a small fraction (~4%) of a Sb^0 (**Fig. R3**). However, it is unlikely in the form of segregated metallic nanocrystals. The key basis is described as follows:

Our XRD and TEM results show no evidence of crystalline Sb^0 phases (**Fig. R4** and **R5**). Crucially, the spatial extent of conductive channels revealed by longitudinal c-AFM significantly surpasses 4% (**Fig. R6**). These results suggests that the limited quantity of Sb^0 atoms does not form a continuous metallic phase but may instead contribute to electronic transport in a more localized manner.

In subsequent analysis, we confirm that some of the Sb^0 can incorporate into the Sb_2O_3 nanocrystal lattice as interstitial defects (I_{Sb}), which may play a critical role in facilitating charge transport across the film, as detailed in our response to **Comment #3**.

Fig. R3 XPS spectrum of $\text{Sb } 3d_{3/2}$ of Sb_2O_3 film.

Fig. R4 TEM images of Sb_2O_3 film.

Fig. R5 XRD spectrum of Sb_2O_3 film.

Fig. R6 AFM and corresponding c-AFM images of the Sb₂O₃ film under a bias voltage of 3 V.

References

- 5 Linarez Pérez, O. E., Sánchez, M. D. & López Teijelo, M. Characterization of growth of anodic antimony oxide films by ellipsometry and XPS. *Journal of Electroanalytical Chemistry* **645**, 143-148 (2010). <https://doi.org/10.1016/j.jelechem.2010.04.023>

Comment #3. Based on the current experimental data, the amorphous-nanocrystalline vertical conduction mode in Sb_2O_3 is not entirely convincing. Could the authors provide additional supporting evidence?

Response: Thank you for the valuable comments. We have systematically investigated the conduction mechanism in thermally evaporated Sb_2O_3 films by integrating structural, electrical, and energy-level analyses, which collectively support a defect-mediated transport model through embedded nanocrystals.

Structurally, TEM and XRD confirm that the Sb_2O_3 consists of $\alpha\text{-Sb}_2\text{O}_3$ nanocrystals embedded within an amorphous film without Sb nanocrystals (Fig. R4-5). This provides direct structural evidence for the existence of "nanocrystalline pathways" at the microscopic level.

Electrically, the longitudinal c-AFM image reveals that the current does not flow uniformly but is concentrated at numerous discrete "hotspots" associated with these nanocrystals (Fig. R6). Furthermore, the single-junction solar cells with the structure of ITO/Me-4PACz/Perovskite/ C_{60} / Sb_2O_3 /Al maintains excellent performance across a wide range of Sb_2O_3 thickness from 15 to 25 nm (Fig. R7).

Fig. R7 Statistics of PCE , V_{oc} , J_{sc} and FF of the PSCs with different Sb_2O_3 thicknesses.

Despite being a wide-bandgap material (~ 4.25 eV, Fig. R8) that creates a substantial interfacial energy

barrier of ~ 1.2 eV at the C_{60}/Sb_2O_3 interface (as revealed by UPS, **Fig. R9**), Sb_2O_3 still allows efficient electron injection across thick layers. Besides, UV-Vis spectra of 100 nm Sb_2O_3 show significant sub-bandgap absorption, indicating abundant in-gap states (**Fig. R10**). These results rule out quantum tunneling⁵ or homogeneous amorphous conduction⁶, possibly indicating an **efficient defect-assisted and nanocrystal-mediated percolative transport mode**.

Fig. R8 Tauc-plot of Sb_2O_3 film.

Fig. R9 UPS spectra of C₆₀ and Sb₂O₃ films.

Fig. R10 UV-vis absorption spectra of 100 nm Sb₂O₃ films under different processing conditions: (i) as-deposited, (ii) after annealing in N₂ at 350 °C, and (iii) after annealing in air at 350 °C; The difference in absorption spectrum (air-annealed minus as-deposited samples).

We further performed a series of correlated experiments and calculations to verify our hypothesis. We conducted a controlled annealing experiment (**Fig. R11**): the 100 nm Sb₂O₃ film was annealed at 350 °C under N₂ atmosphere, which resulted in negligible change in its sub-bandgap absorption. In contrast, the Sb₂O₃ film annealed in air dramatically suppressed the sub-bandgap absorption. This distinct response identifies oxygen-related defects (including interstitial Sb, I_{Sb}) as the primary source of the sub-gap states. Meanwhile, XPS confirmed the presence of a reduced Sb species (**Fig. R3**), originating from I_{Sb} (detailed in our response to **Comment #2**)⁷. We further calculated the density of states (DOS) for various defect types (**Fig. R11-12**). DFT calculations further identify I_{Sb}-induced donor (E_D of 2.90 eV) and acceptor levels (E_{A1} of 1.06 eV and E_{A2} of 1.86 eV) within the gap (**Fig. R13**), with the donor level aligned closely with the C₆₀ LUMO, enabling efficient defect-assisted injection. In contrast, oxygen vacancy-related states were less favorably positioned, pointing to

I_{sb} as the dominant functional defect.

To bridge theory calculation and experiment, we correlate the measured absorption difference spectrum ($\Delta\alpha$, the difference between the two absorption spectra) with the defect energy levels identified by theoretical calculations (**Fig. R10** and **R13**). The inflection points in the $\Delta\alpha$ spectrum at 2.2, 2.74, and 3.27 eV correspond to optical transitions from EA_2 to CBM, from VBM to E_D , and from EA_1 to CBM, respectively. The nearly linear evolution of the absorption difference between 2.2 and 3.27 eV signifies the formation of quasi-continuous defect state around these I_{sb} . This band effectively bridges the ~ 1.2 eV interface energy barrier, enabling efficient electron transport from the LUMO of C_{60} via a defect-assisted percolation mechanism (**Fig. R14**). The merging of discrete theoretical defect levels into a quasi-continuous band, as observed experimentally, results from inhomogeneous broadening and state hybridization induced by high defect densities, local micro-environment fluctuations, and defect-defect interactions within the realistic film.

Fig. R11 Electronic band structures of perfect cubic Sb_2O_3 and defective systems with various point defects calculated by DFT (e.g., Sb interstitials or oxygen vacancies).

Fig. R12 Top views of the Sb_2O_3 crystal and defect structures (e.g. I_{Sb} and V_{O}).

Fig. R13 Energy level structure and optical transitions induced by I_{Sb} defects. Schematic diagram of the Sb_2O_3 energy levels upon introduction of I_{Sb} , obtained from DFT calculations. The optical transition energies E_1 and E_2 from the acceptor levels E_{A1} and E_{A2} to the CBM, and E_3 from the VBM to the donor level E_D are indicated.

Fig. R14 Schematic energy level alignment and electron transport mechanism at the C₆₀/Sb₂O₃ interface. The energy level diagram of C₆₀ and Sb₂O₃, based on UPS results, illustrates that electrons can be efficiently transported from the LUMO of C₆₀ to the Sb₂O₃ layer via a quasi-continuous defect band induced by I_{Sb}, despite the large conduction band offset (~1.2 eV).

To further evaluate this transport mechanism, TPC/TPV analyses were performed on Sb₂O₃- and SnO_x-based devices (**Fig. R15-16**). The TPC decay dynamics show comparable lifetimes, confirming efficient electron extraction via defect-assisted transport in Sb₂O₃. Meanwhile, the matching TPV decays indicate that these defect states do not introduce severe non-radiative recombination.

In conclusion, Sb₂O₃ enables efficient vertical charge transport and extraction via defect-assisted conduction through its nanocrystalline pathways. This mechanism successfully overcomes the high energy

barrier at the C_{60}/Sb_2O_3 interface, providing a new material strategy for developing high-performance of devices.

Fig. R15 TPC measurements of SnO_x - and Sb_2O_3 -based PSCs.

Fig. R16 TPV measurements of SnO_x - and Sb_2O_3 -based PSCs.

References

- 5 Linarez Pérez, O. E., Sánchez, M. D. & López Teijelo, M. Characterization of growth of anodic antimony oxide films by ellipsometry and XPS. *Journal of Electroanalytical Chemistry* **645**, 143-148 (2010). <https://doi.org/10.1016/j.jelechem.2010.04.023>
- 6 Liu, J. *et al.* Efficient and stable perovskite-silicon tandem solar cells through contact displacement by MgFx. *Science* **377**, 302-306 (2022). <https://doi.org/10.1126/science.abn8910>
- 7 Chen, P. *et al.* Multifunctional ytterbium oxide buffer for perovskite solar cells. *Nature* **625**, 516-522 (2024). <https://doi.org/10.1038/s41586-023-06892-x>

Comment #4. Reducing the fullerene thickness is crucial for minimizing optical parasitic absorption.

However, what is the coverage of a 5 nm C₆₀ layer? Does this affect electron extraction efficiency?

Response: Thank you for the valuable comments. We presented the cross-sectional transmission electron microscopy (TEM) image of the PVK/C₆₀/Sb₂O₃/IZO sample to intuitively demonstrate the higher coverage of a 5 nm C₆₀ (Fig. R17). To further examine the coverage, we deposited a 5 nm C₆₀ film on a 100 cm² large-area substrate. Next, we selected five positions on the C₆₀ film to conduct UV-vis measurement (Fig. R18 and Table R1). The similar results of transmittance reveal the superior coverage uniformity of 5 nm C₆₀ film at the large-area scale. Besides, we fabricated 5 nm C₆₀ films as electron transport layer in 100 sub-cells with the device structure illustrated in Fig. R19, showing uniform PCE distribution.

Fig. R17 Cross-sectional TEM image and the corresponding EDX mapping of PVK/C₆₀/Sb₂O₃/IZO stack structure.

Fig. R18 Photograph of a 5 nm C_{60} on a $10 \times 10 \text{ cm}^2$ glass substrate and transmittance spectra measured from five representative positions.

Fig. R19 Statistical distribution of normalized PCE with 100 sub-cells based on Sb_2O_3 on a $10 \times 10 \text{ cm}^2$ substrate.

Meanwhile, we further investigated the potential influence of C_{60} thickness on electron extraction. We demonstrated slightly different energy level distributions and carrier lifetimes between the 5 and 15 nm C_{60} (**Fig. R20-22** and **Table R2**), which implies their comparable electronic driving forces and extraction efficiencies. Importantly, we provided favorable photovoltaic performance data of single-junction device with 5 nm C_{60} and 15nm Sb_2O_3 , confirming the excellent extraction ability of 5 nm C_{60} (**Fig. R23**).

Fig. R20 UPS spectra of 5 and 15 nm C_{60} .

Fig. R21 Energy level scheme of 5 and 15 nm C_{60} .

Fig. R22 TRPL decay curves of PVK/C₆₀-15 and PVK/C₆₀-5 samples.

Fig. R23 The reverse-scan J-V curves of the 1.64 eV-bandgap champion PSCs with SnO_x or Sb₂O₃.

Table R1 The average transmittance of 5 nm C₆₀ film in the wavelength range from 300 nm to 560 nm measured from five representative positions.

	Position 1	Position 2	Position 3	Position 4	Position 5
Average transmittance (%)	81.50	81.78	81.84	81.97	82.08

Table R2 Extracted carrier lifetimes from TRPL measurements. The decay was modeled using a bi-exponential function, showing the fast (τ_1) and slow (τ_2) components, their amplitudes (A_1 and A_2).

	τ_1 (ns)	A_1	τ_2 (ns)	A_2
PVK/C ₆₀ -5	4.41	1205	16.07	235
PVK/C ₆₀ -15	3.93	1742	25.34	182

Comment #5. Figure S8 contains a clear error; please review it thoroughly. Additionally, the valence band edge truncation in Figure S6 appears questionable. It is recommended to increase the number of experimental samples to ensure statistically robust results.

Response: Thank you for pointing out these issues regarding **Fig. S8** and **S6**. We sincerely apologize for the error in **Fig. S8** and any confusion it may have caused. We have conducted a systematic and thorough review of all supplementary figures. We have identified and corrected the specific error in **Fig. S8** and have replaced it with the revised version. The detailed analysis and calibration process are as follows:

The energy level structure of the Sb_2O_3 has been established by combining DFT calculations with consistent results from multiple experimental characterizations, e.g. TEM, XRD. The electronic band structure of cubic Sb_2O_3 was investigated using DFT calculations, which yielded a direct band gap of 4.20 eV (**Fig. R11-12**). This theoretical value is well aligned with the optical band gap of approximately 4.25 eV determined from our UV-Vis measurements (**Fig. R8**), thereby validating both the computational model and the accuracy of UV-vis data. To further ensure the accuracy of the energy level diagram, UPS measurements were meticulously repeated and cross-verified at three independent institutions (**Fig. R24**). The high consistent results confirm the reliability of the energy level structure of Sb_2O_3 , as shown in **Fig. R25**.

Fig. R24 UPS spectra of Sb_2O_3 films from different test institutions.

Fig. R25 Energy level scheme of Sb_2O_3 films from different test institutions.

Comment #6. Since the primary advantage of Sb_2O_3 over SnO_x in this study is the reduced chemical etching of perovskite films, the results of various stability tests are critical. Stability data under continuous illumination at maximum power point tracking, temperature, and electrical bias should be included for devices incorporating Sb_2O_3 buffer layers, along with appropriate explanations.

Response: Thank you for the valuable comments. We fully agree that long-term device stability is indispensable for practical applications. Thus, we systematically evaluated the **thermal (65°C, N_2)** and **optical stability (100 mW/cm² white LED illumination, N_2)** of intrinsic Sb_2O_3 , Sb_2O_3 -based stack films and devices across different time dimensions.

For the intrinsic Sb_2O_3 films, TEM and XRD results revealed no structural changes after thermal or optical aging, demonstrating its robust microstructural stability (Fig. R26-27). XPS analysis demonstrated nearly consistent area of the I_{Sb} -related peak before and after aging, indicating excellent chemical stability of the key defect species (Fig. R28).

Fig. R26 TEM images of Sb_2O_3 films under different aging conditions. (i) Pristine film, and films aged after 40 days in N_2 atmosphere under (ii) 65 °C or (iii) a white LED source (100 mW/cm²).

Fig. R27 GIXRD patterns ($\theta_{inc}=0.3, 1$ and 3°) of Glass/C₆₀/Sb₂O₃ films under different aging conditions. (i) Pristine film, and films aged after 40 days in N₂ atmosphere under (ii) 65 °C or (iii) a white LED source (100 mW/cm²).

Fig. R28 XPS spectra of Sb₂O₃ films under different aging conditions. (i) Pristine film, and films aged after 40 days in N₂ atmosphere under (ii) 65 °C or (iii) a white LED source (100 mW/cm²).

For Sb₂O₃-based stack films, optical microscopy images of C₆₀/Sb₂O₃ samples before and after aging

show no interlayer cracks, warps, delamination, exhibiting excellent stress stability (**Fig. R29**). The GIXRD measurement further confirmed negligible stress gradient within the Sb_2O_3 film and poor residual stress at the $\text{C}_{60}/\text{Sb}_2\text{O}_3$ interface⁸ (**Fig. R27**), attribute to simple physical interaction via van der Waals. Moreover, aging studies of $\text{PVK}/\text{C}_{60}/\text{SnO}_x$ or $\text{Sb}_2\text{O}_3/\text{IZO}$ structures demonstrate the excellent stability of both configurations (**Fig. R30**). SEM and XRD analyses confirm well-maintained structural integrity with no observable adverse effects of Sb_2O_3 on the perovskite layer (**Fig. R31-33**)

Fig. R29 Optical microscopy images of $\text{Glass}/\text{C}_{60}/\text{Sb}_2\text{O}_3$ samples before and after thermal, light aged tests.

Fig. R30 Photographs of $\text{Glass}/\text{Me-4PACz}/\text{PVK}/\text{C}_{60}/\text{SnO}_x$ - or $\text{Sb}_2\text{O}_3/\text{patterned IZO}$ stacks under different aging conditions. (i) Pristine film, and films aged after different days in N_2 atmosphere under (ii) 65°C or (iii) a white LED source ($100\text{ mW}/\text{cm}^2$). (Note: The sample areas were reduced after 40 days due to sample sectioning for characterization.)

Fig. R31 SEM images of PVK/C₆₀/SnO_x or Sb₂O₃/IZO samples before and after thermal aging at 65°C in a N₂ atmosphere.

Fig. R32 SEM images of PVK/C₆₀/SnO_x or Sb₂O₃/IZO samples before and after light aging under a white LED lamp illumination at 100 mW/cm² in a N₂ atmosphere.

Fig .R33 XRD patterns of Glass/Me-4PACz/PVK/C₆₀/SnO_x- or Sb₂O₃/IZO stacks under different aging conditions. (i) Pristine film, and films aged after different days in N₂ atmosphere under (ii) 65 °C or (iii) a white LED source (100 mW/cm²).

For **encapsulated** full devices, Sb₂O₃-based PST were subjected to 500 hours of MPP tracking under white LED lamp illumination at 100 mW/cm² (**Fig. R34**). The device maintained their initial *PCE* without substantial loss. In addition, we continuously monitored the light and thermal stability for more than 1,000 hours under light (white LED lamp illumination at 100 mW/cm²) and 65 °C thermal aging tests, respectively (**Fig. R35-36**).

Both encapsulated devices demonstrated robust photothermal stability, retaining over 90% of their initial *PCE*.

Fig. R34 Normalized *PCE*, V_{oc} , J_{sc} , *FF* of encapsulated Sb_2O_3 -based PST for MPP tracking under a white LED lamp illumination at 100 mW/cm^2 in air.

Fig. R35 Normalized *PCE*, V_{oc} , J_{sc} , *FF* of encapsulated SnO_x - or Sb_2O_3 -based tandem devices for thermal aging test at 65°C in a N_2 atmosphere.

Fig. R36 Normalized PCE , V_{OC} , J_{SC} , FF of encapsulated SnO_x- or Sb₂O₃-based tandem devices for light aging tests under a white LED lamp illumination at 100 mW/cm² in a N₂ atmosphere.

References

- 8 Zheng, L. et al. Strain-induced rubidium incorporation into wide-bandgap perovskites reduces photovoltage loss. *Science* 388, 88-95 (2025). <https://doi.org/10.1126/science.adt3417>

Reviewer #2 (Remarks to the Author):

This work reports a promising strategy of replacing ALD-deposited SnO_x layer with an Sb₂O₃ layer to reduce optical losses in the C₆₀/SnO_x stacked layer. This approach reduces the thickness of the C₆₀ layer, decreases the parasitic absorption and finally increases the J_{SC} , and protects the perovskite layer. This strategy is universal to a broad range of perovskite solar cells with various bandgaps and to large-area tandem solar cells. This work is of great interest and significance to the community of perovskite single-junction and tandem solar cells, especially this work provides an alternative thermal evaporation deposition technique to the existing ALD technique, which could reduce the fabrication cost. Therefore, I would recommend publication of this manuscript after addressing the following minor issues.

Comment #1. As is well known, C₆₀ could increase the fabrication cost of perovskite solar cells. Completely substituting C₆₀ seems a great challenge for perovskite solar cells. Could the authors comment on whether Sb₂O₃ can replace C₆₀ and work as an ETL in tandem cells?

Response: Thank you for the valuable comments. To verify whether Sb₂O₃ can directly replace C₆₀ as an ETL, we fabricated single-junction device with the ITO/Me-4PACz/perovskite/Sb₂O₃/Al structure. $J-V$ curve shows a poor PCE only 14.63% (**Fig. R1**), which is attributed to the large R_s of the device. We further modified the perovskite/Sb₂O₃ interface using PDADI, improving the PCE to 16.58%. However, it can be found that the FF of the device is still relatively low, which may be determined by its higher density of interfacial defects and poorer electrical conductivity compared to C₆₀. Currently, Sb₂O₃ cannot be directly used as an ETL in single-junction or tandem devices. Nevertheless, it is feasible to conduct modifications in term of the electrical properties of Sb₂O₃ and the interface defects between perovskite and Sb₂O₃. Thus, Sb₂O₃ material exhibits significant promise as a future ETL, offering a combination of superior optical management and cost-

effectiveness.

Fig. R1 The J - V curves of champion PSCs based on Sb₂O₃ as ETL.

Comment #2. The authors claim that the higher mobility of Sb_2O_3 is attributed to more favorable energy level alignment. Please explain more on this.

Response: Thank you for the valuable comments. In previous work, the conduction mechanism in Sb_2O_3 was misinterpreted due to inaccuracies in the quantification of its energy levels, for which we apologize and take responsibility for the provided data. To rectify this, we have ensured the accuracy of the Sb_2O_3 energy level alignment through cross-verification at three independent institutions (**Fig. R2-4**). The corrected energy level diagram reveals a ~ 1.2 eV energy barrier at the $\text{C}_{60}/\text{Sb}_2\text{O}_3$ interface (**Fig. R5**), in principle, completely suppress electron injection. Nevertheless, Sb_2O_3 -based single-junction devices achieve high efficiency that is insensitive to its thickness (**Fig. R6**), confirming substantial electron extraction and transport. UV-Vis spectroscopy indicates a high density of in-gap states within 100 nm Sb_2O_3 films (**Fig. R7**). Collectively, these findings point to a unique defect-mediated conduction mechanism distinct from quantum tunneling.

Fig. R2 Tauc-plot of Sb_2O_3 film.

Fig. R3 UPS spectra of Sb_2O_3 films from different test institutions.

Fig. R4 Energy level scheme of Sb_2O_3 films from different test institutions.

Fig. R5 UPS spectra of C_{60} and Sb_2O_3 films.

Fig. R6 Statistics of PCE , V_{oc} , J_{sc} and FF of the PSCs with different Sb_2O_3 thicknesses.

Fig. R7 UV-vis absorption spectra of 100 nm Sb_2O_3 films under different processing conditions: (i) as-deposited, (ii) after annealing in N_2 at 350 °C, and (iii) after annealing in air at 350 °C; The difference in absorption spectrum (air-annealed minus as-deposited samples).

We further performed a series of correlated experiments and calculations to verify our hypothesis. We conducted a controlled annealing experiment (**Fig. R5**): the 100 nm Sb_2O_3 film was annealed at 350 °C under N_2 atmosphere, which resulted in negligible change in its sub-bandgap absorption. In contrast, the Sb_2O_3 film annealed in air dramatically suppressed the sub-bandgap absorption. This distinct response identifies oxygen-related defects (including interstitial Sb, I_{Sb}) as the primary source of the sub-gap states. Meanwhile, XPS confirmed the presence of the Sb^0 atoms⁷ (**Fig. R8**), which serve as precursors for the formation of I_{Sb} . We further calculated the density of states (DOS) for various defect types (**Fig. R9-10**). DFT calculations further identify I_{Sb} -induced donor (E_{D} of 2.90 eV) and acceptor levels (E_{A1} of 1.06 eV and E_{A2} of 1.86 eV) within the gap (**Fig. R11**), with the donor level aligned closely with the C_{60} LUMO, enabling efficient defect-assisted injection. In contrast, oxygen vacancy-related states were less favorably positioned, pointing to I_{Sb} as the dominant functional defect.

To bridge theory calculation and experiment, we correlate the measured absorption difference spectrum ($\Delta\alpha$, the difference between the two absorption spectra) with the defect energy levels identified by theoretical calculations (**Fig. R5** and **R11**). The $\Delta\alpha$ exhibits distinct inflection points at 2.2 and 3.27 eV, which attributed to optical transitions from the E_{A2} and E_{A1} to the CBM, respectively. The inflection point at 2.74 eV is assigned to transitions from the VBM to the E_D . The nearly linear evolution of the absorption difference between 2.2 and 3.27 eV signifies the formation of quasi-continuous defect state around these I_{Sb} . This band effectively bridges the ~ 1.2 eV interface energy barrier, enabling efficient electron transport from the LUMO of C_{60} via a defect-assisted percolation mechanism (**Fig. R13**). Besides, the discrepancy between the observed quasi-continuous defect band and the calculated discrete levels can be understood by considering the realistic film environment. High defect density, local micro-environment fluctuations, and defect-defect interactions cause significant inhomogeneous broadening and state hybridization, ultimately merging discrete levels into a continuous defect band. Sum up, Sb_2O_3 enables efficient vertical charge transport and extraction via defect-assisted conduction through its nanocrystalline pathways.

Fig. R8 XPS spectrum of Sb 3d_{3/2} of Sb₂O₃ film.

Fig. R9 Electronic band structures of perfect cubic Sb_2O_3 and defective systems with various point defects calculated by DFT (e.g., Sb interstitials or oxygen vacancies).

Fig. R10 Top views of the Sb_2O_3 crystal and defect structures (e.g. I_{Sb} and V_{O}).

Fig. R11 Energy level structure and optical transitions induced by I_{Sb} defects. Schematic diagram of the Sb_2O_3 energy levels upon introduction of I_{Sb} , obtained from DFT calculations. The optical transition energies E_1 and E_2 from the acceptor levels E_{A1} and E_{A2} to the CBM, and E_3 from the VBM to the donor level E_D are indicated.

Fig. R12 Schematic energy level alignment and electron transport mechanism at the C_{60}/Sb_2O_3 interface. The energy level diagram of C_{60} and Sb_2O_3 , based on UPS results, illustrates that electrons can be efficiently transported from the LUMO of C_{60} to the Sb_2O_3 layer via a quasi-continuous defect band induced by I_{Sb} , despite the large conduction band offset (~ 1.2 eV).

To further evaluate the efficacy of this transport mechanism, we conducted comparative TPC and TPV measurements on Sb_2O_3 - and SnO_x -based devices (**Fig. R13-14**). The TPC results showed comparable photocurrent decay lifetimes, demonstrating that the defect-assisted transport in Sb_2O_3 enables efficient electron extraction. Simultaneously, the TPV measurements revealed similar photovoltage decay lifetimes, indicating that the defect states in Sb_2O_3 do not introduce severe non-radiative recombination losses. Thus, these findings confirm that the defect-assisted transport mechanism achieves high-efficiency charge collection while

maintaining high interfacial quality.

In conclusion, Sb_2O_3 enables efficient vertical charge transport and extraction via defect-assisted conduction through its nanocrystalline pathways. This mechanism successfully overcomes the high energy barrier at the $\text{C}_{60}/\text{Sb}_2\text{O}_3$ interface, providing a new material strategy for developing high-performance of devices.

Fig. R13 TPC measurements of SnO_x - and Sb_2O_3 -based PSCs.

Fig. R14 TPV measurements of SnO_x - and Sb_2O_3 -based PSCs.

References

- 5 Linarez Pérez, O. E., Sánchez, M. D. & López Tejelo, M. Characterization of growth of anodic antimony oxide films by ellipsometry and XPS. *Journal of Electroanalytical Chemistry* **645**, 143-148 (2010). <https://doi.org/10.1016/j.jelechem.2010.04.023>

Comment #3. In Figure 2d and 2f, please clarify the voltage scan direction. In Figure 2h, please clarify what the current densities correspond to.

Response: Thank you for the valuable comments and have updated the text and figure accordingly. We clarified that the J - V curves in **Fig. 2c** and **2e** (previous **Fig. 2d** and **2f**) were both measured in reverse scan from 1.3 to 0 V at a rate of 0.2 V/s. As shown in **Fig. S24** (previous **Fig. 2h**), the integrated J_{SC} from the EQE spectra of PSCs with 1.59, 1.62, 1.64 and 1.68 eV bandgaps matched well with the J_{SC} values from J - V curves (**Fig. 2c** and **2e**).

Comment #4. Section 5 of Page 10: to examine the homogeneity of Sb_2O_3 , the authors deposited a 45 nm Sb_2O_3 layer on $10\times 10\text{ cm}^2$ glass substrate. Could the authors comment on uniformity of the 15 nm thick Sb_2O_3 for scaling up cells?

Response: Thank you for the valuable comments. Following your suggestions, we deposited a 15 nm Sb_2O_3 film on 100 cm^2 substrate. We selected five representative positions on the film and conducted the AFM tests (**Fig. R15-16**), showing the approximate mean step heights ($\sim 15\text{ nm}$) of the five sub-samples. Moreover, the similar surface potential ($\sim 700\text{ mV}$) of these five sub-samples further confirmed the superior uniformity of the 15 nm Sb_2O_3 film (**Fig. R17**). Besides, we fabricated 5nm C_{60} films as electron transport layer in 100 sub-cells with the device structure illustrated in **Fig. R18**, showing uniform *PCE* distribution.

Fig. R15 Photograph of a 15 nm Sb_2O_3 on a $10\times 10\text{ cm}^2$ glass substrate and transmittance spectra measured from five representative positions.

Fig. R16 AFM images and mean step height of 15 nm Sb_2O_3 films measured from five positions.

Fig. R17 AFM and KPFM images of 15 nm Sb_2O_3 films measured from five positions.

Fig. R18 Statistical distribution of normalized PCE with 100 sub-cells based on Sb_2O_3 on a $10 \times 10 \text{ cm}^2$ substrate.

Comment #5. Page 8: to evaluate the optical benefits of Sb_2O_3 layer and reduced C_{60} thickness, the authors should compare the average transmittance of $\text{C}_{60}/\text{Sb}_2\text{O}_3$ and $\text{C}_{60}/\text{SnO}_x$ stacked layers.

Response: Thank you for the valuable comments. To explicitly illustrate the optical benefits afforded by Sb_2O_3 , we re-implement UV-vis spectroscopy measurements on both the $\text{C}_{60}\text{-5}/\text{Sb}_2\text{O}_3$ and $\text{C}_{60}\text{-15}/\text{SnO}_x$ samples, respectively in **Fig. R19** (**Fig. 3g** in **Manuscript**). Furthermore, the average transmittance (T_{avg}) of the $\text{C}_{60}\text{-5}/\text{Sb}_2\text{O}_3$ (74.85%) and $\text{C}_{60}\text{-15}/\text{SnO}_x$ (62.68%) were quantified in the wavelength range from 300 to 560 nm and presented in the corresponding **Fig. 3g**.

Fig. R19 Absorbance and transmittance spectrum of $\text{C}_{60}/\text{Sb}_2\text{O}_3$ and SnO_x films.

Reviewer #3 (Remarks to the Author):

Shi et al. introduce an antimony oxide as buffer layer for perovskite based single junction and tandem solar cells. This approach enhances the power conversion efficiency (PCE) and stability of perovskite/silicon tandem cells. The optimized tandem device achieves a PCE of 30.1%. These results underscore the strategy's potential for industrial-scale applications. However, the stability and efficiency of the devices in the article is not that impressive to guarantee its publication in Nature Communications. Overall, this work demonstrates interesting result with good material design. Therefore, I recommend this paper to be reconsidered after revisions and suggest some points for further improvement.

Comment #1. The maximum power point testing should be incorporated in the manuscript with certain duration. Additionally, post-aging microstructural analysis using SEM, XRD and TOF-SIMS should be applied correspondingly.

Response: Thank you for the valuable comments. Combined with **Comment #9** and **#10**, we performed comprehensive aging researches on intrinsic Sb_2O_3 films, multilayer stacks, and complete devices. It is crucial to clarify that in the application context of this study, Sb_2O_3 serves as a buffer layer within an encapsulated device. Since effective encapsulation mitigates the ingress of moisture and oxygen, our investigation targets the **optical (100 mW/cm² white LED illumination) and thermal (65°C) stability** in a N_2 atmosphere for 40 days. (Besides, because it was impossible to extract the encapsulated PSTs for direct measurement, we designed PVK/C₆₀/SnO_x or Sb_2O_3 /IZO stacks to simulate the actual device environment, allowing for post-aging analysis.)

First, the intrinsic stability of Sb_2O_3 was evaluated through multiple characterization techniques. Both TEM and XRD results confirmed the crystal structural integrity of the Sb_2O_3 after aging (**Fig. R1-2**). XPS spectra showed no obvious change of the peak information before and after aging, indicating its remarkable chemical

stability (Fig. R3).

Fig. R1 TEM images of the pristine, thermal aged and light aged Sb_2O_3 films. All aging tests were conducted in a N_2 atmosphere for 40 days. The thermal aging was performed at 65°C , and the optical aging used a white LED source (100 mW/cm^2).

Fig. R2 GIXRD patterns ($\theta_{\text{inc}}=0.3, 1$ and 3°) of $\text{Glass/C}_{60}/\text{Sb}_2\text{O}_3$ films under different aging conditions. (i) Pristine film, and films aged after 40 days in N_2 atmosphere under (ii) 65°C or (iii) a white LED source (100 mW/cm^2).

Fig. R3 XPS spectra of the pristine, thermal aged and light aged Sb_2O_3 films.

Second, the Sb_2O_3 -based stack architectures demonstrated excellent interfacial stability under aging. Optical microscopy confirmed that the $\text{C}_{60}/\text{Sb}_2\text{O}_3$ interface, governed by van der Waals interaction, exhibited no cracking, warping, or delamination, indicating effective stress release. (**Fig. R4**). Moreover, the microstructure of the $\text{C}_{60}/\text{Sb}_2\text{O}_3$ sample was examined by GIXRD in **Fig. R2**. By tuning the incident angle θ_{inc} from 0.3° to 3° , the probe depth was varied from the surface to the deeper regions of the film⁸, which was confirmed by the emergence of a broad diffraction signal from the amorphous glass/ C_{60} substrate at $\theta_{\text{inc}}=3^\circ$. Notably, XRD peaks showed no shift with depth before and after aging, confirming negligible stress gradient within the Sb_2O_3 film and poor residual interfacial stress, attributed to the van der Waals contact at the $\text{C}_{60}/\text{Sb}_2\text{O}_3$. Moreover, we evaluated the thermal and optical stability of the PVK/ SnO_x/IZO and PVK/ $\text{Sb}_2\text{O}_3/\text{IZO}$ stack, respectively. No noticeable decomposition or morphological degradation was observed by visual inspection or SEM (**Fig. R5-7**). XRD further indicated the aged perovskite nearly maintain intact crystal structure (**Fig. R8**).

Fig. R4 Optical microscopy images of Glass/C₆₀/Sb₂O₃ samples before and after thermal, light aged tests.

Fig. R5 Photographs of Glass/Me-4PACz/PVK/C₆₀/SnO_x- or Sb₂O₃/patterned IZO stacks under different aging conditions. (i) Pristine film, and films aged after different days in N₂ atmosphere under (ii) 65 °C or (iii) a white LED source (100 mW/cm²). (Note: The sample areas were reduced after 40 days due to sample sectioning for characterization.)

Fig. R6 SEM images of PVK/C₆₀/SnO_x or Sb₂O₃/IZO samples before and after thermal aging at 65°C in a N₂ atmosphere.

Fig. R7 SEM images of PVK/C₆₀/SnO_x or Sb₂O₃/IZO samples before and after light aging under a white LED lamp illumination at 100 mW/cm² in a N₂ atmosphere.

Fig .R8 XRD patterns of Glass/Me-4PACz/PVK/C₆₀/SnO_x- or Sb₂O₃/IZO stacks under different aging conditions. (i) Pristine film, and films aged after different days in N₂ atmosphere under (ii) 65 °C or (iii) a white LED source (100 mW/cm²).

Finally, we conducted a MPP tracking of encapsulated Sb₂O₃-based PST for 500 hours under white LED lamp illumination at 100 mW/cm² (**Fig. R9**). The *PCE* of the devices was maintained without significant loss. Besides, the encapsulated SnO_x- and Sb₂O₃-based PST exhibited robust optical and thermal stability during continuous testing exceeding 1,000 hours under a white LED illumination (100 mW/cm²) and thermal aging at

65°C, respectively, retaining over 90% of their initial *PCE* (Fig. R10-11).

Fig. R9 Normalized *PCE*, V_{OC} , J_{SC} , *FF* of encapsulated Sb_2O_3 -based PST for MPP tracking under a white LED lamp illumination at 100 mW/cm^2 in air.

Fig. R10 Normalized *PCE*, V_{OC} , J_{SC} , *FF* of encapsulated SnO_x - or Sb_2O_3 -based tandem devices for thermal aging tests at 65°C in a N_2 atmosphere.

Fig. R11 Normalized PCE , V_{OC} , J_{SC} , FF of encapsulated SnO_x - or Sb_2O_3 -based tandem devices for light aging tests under a white LED lamp illumination at 100 mW/cm^2 in a N_2 atmosphere.

References

- 8 Zheng, L. et al. Strain-induced rubidium incorporation into wide-bandgap perovskites reduces photovoltage loss. *Science* 388, 88-95 (2025). <https://doi.org/10.1126/science.adt3417>

Comment #2. The TEM image in Fig. 1g appears blurred, and the interfaces are difficult to distinguish. A low-magnification TEM image should be provided to clearly demonstrate the long-range behavior of the interlayer.

Response: Thank you for the valuable comments. To show the long-range behavior of the interlayer, we re-prepared the cross-sectional sample with the structure of Si/NiO_x/Me-4PACz/PVK/C₆₀/Sb₂O₃/IZO using focused ion beam (FIB) technology and provided its TEM and EDS images at low magnification (Fig. R12-13). The EDS mapping clearly outlines the sandwich structure, in which an inconspicuous amount of indium elements was detected in the C₆₀ and perovskite layers, displaying the qualified sputtering tolerance of the Sb₂O₃ film. Also, the undamaged Sb₂O₃ nanocrystalline with a cubic structure also further verifies its excellent sputtering tolerance.

Fig. R12 Cross-sectional TEM image and EDX mapping of PVK/C₆₀/Sb₂O₃/IZO stack films.

Fig. R13 Cross-sectional TEM image and EDX mapping of Si/NiO_x/Me-4PACz/PVK/C₆₀/Sb₂O₃/IZO stack.

Comment #3. The AFM results show that the RMS roughness decreases after Sb_2O_3 deposition compared to the reference film. Does this imply that Sb_2O_3 does not form a uniform coverage?

Response: Thank you for the valuable comments. In Fig. R14 (SI in Fig. S2), the reduction in surface roughness of the PVK/ C_{60} sample after Sb_2O_3 deposition is most likely associated with to the slightly different thickness of thermally evaporated Sb_2O_3 layers at the valleys and protrusions of the underlying PVK/ C_{60} substrate. The thermally evaporated Sb_2O_3 layer is beneficial to filling the valleys and hence slightly thicker at valleys, thereby contributing to a smoother surface morphology. However, the thermally evaporated Sb_2O_3 layer is uniform enough to cover the rough PVK/ C_{60} substrate.

To further investigate the uniformity of evaporated Sb_2O_3 , we deposited a 15 nm Sb_2O_3 film on a 100 cm^2 substrate. Additionally, we selected five representative positions on this film and conducted thickness measurements on via AFM and KPFM tests (Fig. R15-17). By comprehensively analyzing the data, we confirmed the coverage uniformity of 15 nm Sb_2O_3 . Besides, the uniformity of 15 nm Sb_2O_3 can also be intuitively demonstrated in cross-sectional TEM images (Fig. R12) shown in Comments 2.

Fig. R14 AFM images of PVK/ C_{60} /without and with Sb_2O_3 .

Fig. R15 Photograph of a 15 nm Sb_2O_3 on a $10 \times 10 \text{ cm}^2$ glass substrate and transmittance spectra measured from five representative positions.

Fig. R16 AFM images and mean step height of 15 nm Sb_2O_3 films measured from five positions.

Fig. R17 AFM and KPFM images of 15 nm Sb_2O_3 films measured from five positions.

Comment #4. The explanation of the electron transfer mechanism remains unclear. Additional characterizations are necessary to elucidate the underlying charge transfer behavior in the Sb_2O_3 layer.

Response: Thank you for the valuable comments. Through systematic investigation across structural, electrical, and energetic dimensions, we have confirmed the existence of a **defect-mediated nanocrystal transport mechanism** in thermally evaporated Sb_2O_3 films.

Structurally, TEM and XRD results verify the embedding of $\alpha\text{-Sb}_2\text{O}_3$ nanocrystals within an amorphous matrix, establishing the nanocrystal conduction pathways (**Fig. R18-19**).

Fig. R18 TEM images of Sb_2O_3 film.

Fig. R19 XRD spectrum of Sb_2O_3 film.

Electrically, longitudinal c-AFM image reveals current localization at nanocrystalline hotspots, indicating the electron transport by the nanocrystals as channel (Fig. R20). Meanwhile, the Sb₂O₃-based single-junction devices exhibit high-efficiency and stable performance across 15-25 nm Sb₂O₃ (Fig. R21).

Fig. R20 AFM and corresponding c-AFM images of the Sb₂O₃ film under a bias voltage of 3 V.

Fig. R21 Statistics of *PCE*, *V_{oc}*, *J_{sc}* and *FF* of the PSCs with different Sb₂O₃ thicknesses.

Energetically, as a wide-bandgap material (~4.25 eV), Sb₂O₃ tends to form high energy barriers at interface (Fig. R22). The UPS results reveal a substantial interfacial energy barrier of ~1.2 eV at the C₆₀/Sb₂O₃ interface (Fig. R23). Among, the UPS results of Sb₂O₃ were reproduced and cross-verified at three independent

institutions, ensuring its accuracy (Fig. R24-25). But the efficient electron injection persists even through thick Sb_2O_3 layers (Fig. R21), inconsistent with quantum tunneling⁵ or amorphous conduction⁶. Furthermore, UV-vis spectra of 100 nm Sb_2O_3 film shows significant sub-bandgap absorption (Fig. R26), indicating the existence of abundant gap states. Based on these data, we support a **defect-assisted, nanocrystal-mediated percolation transport model**.

Fig. R22 Tauc-plot of Sb_2O_3 film.

Fig. R23 UPS spectra of C_{60} and Sb_2O_3 films.

Fig. R24 UPS spectra of Sb_2O_3 films from different test institutions.

Fig. R25 Energy level scheme of Sb_2O_3 films from different test institutions.

Fig. R26 UV-vis absorption spectra of 100 nm Sb_2O_3 films under different processing conditions: (i) as-deposited, (ii) after annealing in N_2 at 350 °C, and (iii) after annealing in air at 350 °C; The difference in absorption spectrum (air-annealed minus as-deposited samples).

Next, we systematically verified our hypothesis through correlated experiments and calculations. We conducted a controlled annealing experiment (**Fig. R26**): the 100 nm Sb_2O_3 film was annealed at 350 °C under N_2 atmosphere, which resulted in negligible change in its sub-bandgap absorption. In contrast, the Sb_2O_3 film annealed in air dramatically suppressed the sub-bandgap absorption, indicating oxygen-related defects (e.g. interstitial Sb, I_{Sb}) as the source. Also, XPS confirmed the presence of the Sb^0 atoms (**Fig. R27**), which serve as precursors for the formation of I_{Sb} . To further identify the dominant defect species, we systematically investigated the band structure of $\alpha\text{-Sb}_2\text{O}_3$ using DFT calculations, with particular focus on the defect energy levels introduced by various types of defects (**Fig. R28-29**). Crucially, we observed that the I_{Sb} defects introduce specific in-gap states: two acceptor levels (E_{A1} at 1.06 eV, E_{A2} at 1.86 eV) and a donor level (E_{D} at 2.90 eV). The close alignment of E_{D} with the C_{60} LUMO enables efficient defect-mediated electron injection. In contrast, oxygen vacancy-related states were less favorably positioned, pointing to I_{Sb} as the dominant functional defect.

Systematic correlation of measured absorption difference spectrum ($\Delta\alpha$, the difference between the two absorption spectra) with theoretical calculations provides compelling evidence for defect-mediated charge transport (**Fig. R26**). The $\Delta\alpha$ reveals inflection points at 2.2, 2.74, and 3.27 eV, corresponding to calculated $E_{A2}\rightarrow\text{CBM}$ ($E_2=2.18$ eV), $\text{VBM}\rightarrow E_D$ ($E_3=2.90$ eV), and $E_{A1}\rightarrow\text{CBM}$ ($E_1=2.98$ eV) transitions in whole $\text{VBM}\rightarrow\text{CBM}$ transition, respectively (**Fig. R30**). Besides, the nearly linear absorption evolution between 2.2 and 3.27 eV demonstrates the formation of quasi-continuous defect state around these I_{Sb} levels (**Fig. R26**). This band architecture enables efficient electron injection across the 1.2 eV $\text{C}_{60}/\text{Sb}_2\text{O}_3$ interface barrier via defect-assisted percolation (**Fig. R30**). The transition from discrete theoretical levels to continuous experimental bands is rationally explained by collective environmental effects in real material systems, where high defect density and local fluctuations induce significant state broadening and hybridization.

Fig. R27 XPS spectrum of Sb 3d_{3/2} of Sb₂O₃ film.

Fig. R28 Electronic band structures of perfect cubic Sb_2O_3 and defective systems with various point defects calculated by DFT (e.g., Sb interstitials or oxygen vacancies).

Fig. R29 Top views of the Sb_2O_3 crystal and defect structures (e.g. I_{Sb} and V_{O}).

Fig. R30 Energy level structure and optical transitions induced by I_{Sb} defects. Schematic diagram of the Sb_2O_3 energy levels upon introduction of I_{Sb} , obtained from DFT calculations. The optical transition energies E_1 and E_2 from the acceptor levels E_{A1} and E_{A2} to the CBM, and E_3 from the VBM to the donor level E_D are indicated.

Fig. R31 Schematic energy level alignment and electron transport mechanism at the C_{60}/Sb_2O_3 interface. The energy level diagram of C_{60} and Sb_2O_3 , based on UPS results, illustrates that electrons can be efficiently transported from the LUMO of C_{60} to the Sb_2O_3 layer via a quasi-continuous defect band induced by I_{Sb} , despite the large conduction band offset (~ 1.2 eV).

To further evaluate the efficacy of this transport mechanism, we conducted TPC and TPV measurements (**Fig. R32-33**). The Sb_2O_3 -based devices exhibit photocurrent decay lifetimes comparable to their SnO_x -based counterparts, demonstrating efficient charge extraction via the unique defect-assisted transport. Furthermore, the similar photovoltage decay lifetimes in TPV measurements confirm that the gap states in Sb_2O_3 do not introduce significant non-radiative recombination. Above-mentioned results confirm the effectiveness of the defect-mediated conduction mechanism.

In conclusion, Sb_2O_3 enables efficient vertical charge transport and extraction via defect-assisted conduction through its nanocrystalline pathways. This mechanism successfully overcomes the high energy barrier at the $\text{C}_{60}/\text{Sb}_2\text{O}_3$ interface, providing a new material strategy for developing high-performance of devices.

Fig. R32 TPC measurements of SnO_x - and Sb_2O_3 -based PSCs.

Fig. R33 TPV measurements of SnO_x - and Sb_2O_3 -based PSCs.

References

- 6 Liu, J. *et al.* Efficient and stable perovskite-silicon tandem solar cells through contact displacement by MgF_x. *Science* **377**, 302-306 (2022). <https://doi.org/10.1126/science.abn8910>
- 7 Chen, P. *et al.* Multifunctional ytterbium oxide buffer for perovskite solar cells. *Nature* **625**, 516-522 (2024). <https://doi.org/10.1038/s41586-023-06892-x>

Comment #5. The nature of the $\text{Sb}_2\text{O}_3/\text{C}_{60}$ interface is not well defined. It should be clarified whether the interface is governed by simple physical contact or involves specific chemical interactions.

Response: Thank you for the valuable comments. Based on XRD results (Fig. R34), the Sb_2O_3 layer is a cubic-phase nanocrystal film with an interplanar spacing of $d = 0.32 \text{ \AA}$ ^{4,9,10}. To investigate their interaction style between C_{60} and Sb_2O_3 , we prepared a cross-sectional sample containing the $\text{C}_{60}/\text{Sb}_2\text{O}_3$ films using FIB technology. Subsequently, we further characterized the samples by TEM (Fig. R35), and showed the unchanged microstructure of the Sb_2O_3 nanocrystal in contact with C_{60} . Moreover, we further performed grazing incidence X-ray diffraction (GIXRD) to observe the difference of $\text{C}_{60}/\text{Sb}_2\text{O}_3/\text{IZO}$ sample in vertical microstructure⁸ (Fig. R36). Furthermore, when the incidence angle θ_{inc} increased to 3° , a broad peak representing amorphous C_{60} substrate was observed in the XRD results. No peak shift is observed at the interface between the Sb_2O_3 and C_{60} layers. Therefore, $\text{C}_{60}/\text{Sb}_2\text{O}_3$ interface involves only a **simple physical contact via van der Waals forces**.

Fig. R34 XRD spectrum of Sb_2O_3 film.

Fig. R35 Cross-sectional TEM image of PVK/C₆₀/Sb₂O₃/IZO films.

Fig. R36 GIXRD pattern of the Sb₂O₃ film at different incidence angles ($\theta_{inc}=0.3, 1$ and 3°).

References

- 4 Liu, K. *et al.* A wafer-scale van der Waals dielectric made from an inorganic molecular crystal film. *Nature Electronics* **4**, 906-913 (2021). <https://doi.org/10.1038/s41928-021-00683-w>
- 8 Zheng, L. *et al.* Strain-induced rubidium incorporation into wide-bandgap perovskites reduces photovoltage loss. *Science* **388**, 88-95 (2025). <https://doi.org/10.1126/science.adt3417>
- 9 Han, W. *et al.* Two-dimensional inorganic molecular crystals. *Nature Communications* **10**, 4728 (2019). <https://doi.org/10.1038/s41467-019-12569-9>
- 10 Tigau, N. *et al.* The influence of heat treatment on the electrical conductivity of antimony trioxide thin films. *Journal of Optoelectronics and Advanced Materials* **5**, 907-912 (2003).

Comment #6. It should be specified whether the XPS and UPS measurements were conducted on the Sb_2O_3 film alone or on the $\text{C}_{60}/\text{Sb}_2\text{O}_3$ stack, considering the limited detection depth of these techniques. Any potential band bending effects at the interface should also be discussed.

Response: Thank you for the valuable comments. We fully acknowledge the importance of considering interfacial band bending. Based on UPS results (**Fig. R23** and **R31**), we have obtained the energy level structures of individual C_{60} and Sb_2O_3 layers. It is noteworthy that both C_{60} and Sb_2O_3 , being n-type semiconductors, exhibit nearly identical Fermi level positions (C_{60} : -4.39 eV, Sb_2O_3 : -4.29 eV). This energy level alignment significantly reduces the driving force for built-in electric field formation at the $\text{C}_{60}/\text{Sb}_2\text{O}_3$ interface, making band bending effects negligible.

Fig. R23 UPS spectra of C_{60} and Sb_2O_3 films.

Fig. R31 Schematic energy level alignment and electron transport mechanism at the C₆₀/Sb₂O₃ interface. The energy level diagram of C₆₀ and Sb₂O₃, based on UPS results, illustrates that electrons can be efficiently transported from the LUMO of C₆₀ to the Sb₂O₃ layer via a quasi-continuous defect band induced by I_{Sb}, despite the large conduction band offset (~1.2 eV).

Comment #7. Although the IZO layer thickness is claimed to exceed 50 nm, it is not clearly visible in the SEM image of the perovskite/IZO film. TOF-SIMS measurements may provide more insightful information to correlate potential IZO damage with changes in the perovskite morphology.

Response: Thank you for the valuable comments, which prompted us to conduct a more thorough and repeated investigation of this specific phenomenon. We would like to clarify that we intentionally adopted thin (~10 nm) IZO layer in SEM images of PVK/IZO samples, whereas IZO thickness in the actual device is 40–50 nm.

Moreover, we repeatedly measured the SEM of PVK/IZO and PVK/Sb₂O₃/IZO samples to investigate the sputtering damage (**Fig. R37**). In newly fabricated PVK/IZO samples, we indeed observed no obvious pinholes, which is inconsistent with the previous SEM results (**Fig. R38**). We conclude that the pinhole structures shown in the original manuscript were not representative but likely resulted from occasional factors during sample preparation. However, **it should be noted that the absence of morphological damage does not rule out electrical damage caused by sputter**. Consistent with extensive literature¹¹⁻¹⁵, the PST without a buffer layer exhibited significantly reduced *PCE* and *FF*, confirming the necessity of a protective interlayer (**Fig. R39**). In addition, the critical role of a buffer with sputtering tolerance is highlighted by the poor performance of BCP-based PST^{14,15}, despite its high *PCE* (>22%) in single-junction configurations (**Fig. R40**). Consequently, the high performance of Sb₂O₃-based tandem devices intuitively demonstrates its superior sputtering tolerance.

Fig. R37 Previous measured SEM images of PVK/C₆₀/IZO and PVK/C₆₀/Sb₂O₃/IZO samples.

Fig. R38 SEM images of PVK/C₆₀/IZO and PVK/C₆₀/Sb₂O₃/IZO samples.

Fig. R39 The J - V curves of champion PST without buffer layer and with BCP, BCP: Ag.

Fig. R40 The J - V curve of champion PSC based on BCP buffer layer.

- 11 Yang, Q. *et al.* Origin of sputter damage during transparent conductive oxide deposition for semitransparent perovskite solar cells. *Journal of Materials Chemistry A* **12**, 14816-14827 (2024). <https://doi.org/10.1039/d3ta06654a>
- 12 Kanda, H. *et al.* Analysis of sputtering damage on I–V curves for perovskite solar cells and simulation with reversed diode model. *The Journal of Physical Chemistry C* **120**, 28441-28447 (2016). <https://doi.org/10.1021/acs.jpcc.6b09219>
- 13 Härtel, M. *et al.* Reducing sputter damage-induced recombination losses during deposition of the transparent front-electrode for monolithic perovskite/silicon tandem solar cells. *Solar Energy Materials and Solar Cells* **252**, 112180 (2023). <https://doi.org/10.1016/j.solmat.2023.112180>
- 14 Liu, K. *et al.* Reducing sputter induced stress and damage for efficient perovskite/silicon tandem solar cells. *Journal of Materials Chemistry A* **10**, 1343-1349 (2022). <https://doi.org/10.1039/d1ta09143c>
- 15 Magliano, E. *et al.* Solution-processed metal-oxide nanoparticles to prevent the sputtering damage in perovskite/silicon tandem solar cells. *ACS Applied Materials & Interfaces* **17**, 17599-17610 (2025). <https://doi.org/10.1021/acsami.5c00090>

Comment #8. The reference device employing a SnO₂ buffer layer shows a *PCE* of only ~28%, which is noticeably lower than previously reported values from the same research group. The reason for this performance decline should be clarified.

Response: Thank you for the valuable comments regarding the performance of our SnO₂-based reference device. Compared to perovskite films prepared by the one-step solution method¹⁶⁻¹⁸, those fabricated via the vacuum-solution hybrid (VSH) method exhibit inferior crystallinity, which consequently lead to a relatively lower *PCE* in tandem devices based on the VSH-processed absorber¹⁹⁻²⁴.

Moreover, the reason for the relatively lower *PCE* (~28%) compared to our previous reports is primarily attributed to a strategic shift in our device architecture, specifically the change from a large-textured silicon substrate to a small-textured one²¹⁻²⁴. While the large-textured PSTs offered marginally superior light-trapping and a higher potential J_{SC} , it posed a significant barrier to the uniform deposition of the self-assembled monolayer (SAM)²⁵⁻²⁷, which is crucial for high-performance and reproducible tandem devices. The results led to inconsistent device performance and hindered scalable-PSTs fabrication²⁸. Therefore, we temporarily adopted the small-textured substrate to prioritize SAM uniformity and process reliability. This modification led to an improvement in V_{OC} , but it also caused a slight reduction in J_{SC} ²⁵. In future work, we will further focus on the application of high-efficiency large-textured devices by addressing the issues derived from large-textured substrates.

References

- 16 Jia, L. *et al.* Efficient perovskite/silicon tandem with asymmetric self-assembly molecule. *Nature* **644**, 912-919 (2025). <https://doi.org/10.1038/s41586-025-09333-z>
- 17 Liu, J. *et al.* Perovskite/silicon tandem solar cells with bilayer interface passivation. *Nature* **635**, 596-603 (2024). <https://doi.org/10.1038/s41586-024-07997-7>
- 18 Wu, W. *et al.* Stable and uniform self-assembled organic diradical molecules for perovskite photovoltaics. *Science* **387**, eadv4551 (2025). <https://doi.org/10.1126/science.adv4551>
- 19 Chin, X. Y. *et al.* Interface passivation for 31.25%-efficient perovskite/silicon tandem solar cells. *Science* **381**, 59-62 (2023). <https://doi.org/10.1126/science.adg0091>
- 20 Er-Raji, O. *et al.* Electron accumulation across the perovskite layer enhances tandem solar cells with textured silicon. *Science*, eadx1745 (2025). <https://doi.org/10.1126/science.adx1745>
- 21 Li, Y. *et al.* CsCl induced efficient fully-textured perovskite/crystalline silicon tandem solar cell. *Nano Energy* **122**, 109285 (2024). <https://doi.org/10.1016/j.nanoen.2024.109285>
- 22 Xu, Q. *et al.* Conductive passivator for efficient monolithic perovskite/silicon tandem solar cell on commercially textured silicon. *Advanced Energy Materials* **12**, 2202404 (2022). <https://doi.org/10.1002/aenm.202202404>
- 23 Xu, Q. *et al.* Diffusible capping layer enabled homogeneous crystallization and component distribution of hybrid sequential deposited perovskite. *Advanced Materials* **36**, 2308692 (2024). <https://doi.org/10.1002/adma.202308692>
- 24 Liu, J. *et al.* Textured perovskite/silicon tandem solar cells achieving over 30% efficiency promoted by 4-fluorobenzylamine hydroiodide. *Nano-Micro Letters* **16**, 189 (2024). <https://doi.org/10.1007/s40820-024-01406-4>

- 25 Xu, L. *et al.* Accurate optical modeling of monolithic perovskite/silicon tandem solar cells and modules on textured silicon substrates. *PRX Energy* **1**, 023005 (2022). <https://doi.org/10.1103/PRXEnergy.1.023005>
- 26 Zhang, X. *et al.* A spiro-type self-assembled hole transporting monolayer for highly efficient and stable inverted perovskite solar cells and modules. *Energy & Environmental Science* **18**, 468-477 (2025). <https://doi.org/10.1039/d4ee01960a>
- 27 Park, S. M. *et al.* Low-loss contacts on textured substrates for inverted perovskite solar cells. *Nature* **624**, 289-294 (2023). <https://doi.org/10.1038/s41586-023-06745-7>
- 28 Zheng, X. *et al.* Co-deposition of hole-selective contact and absorber for improving the processability of perovskite solar cells. *Nature Energy* **8**, 462-472 (2023). <https://doi.org/10.1038/s41560-023-01227->

Comment #9. The intrinsic stability of the Sb_2O_3 layer has not been discussed. Additionally, its potential impact on the long-term stability (under thermal treatment, light illumination, and moisture) of the underlying perovskite film remains unclear and should be addressed.

Response: Thank you for the valuable comments. We fully agree with the importance of the intrinsic stability of Sb_2O_3 film and its potential impact on the stability of perovskite. As detailed in our response to **Comment #1**, we have demonstrated that intrinsic Sb_2O_3 exhibits excellent optical and thermal stability. After aging tests, no significant changes were observed in either its physical structure or chemical state (**Fig. R1-3**). Besides, the $\text{C}_{60}/\text{Sb}_2\text{O}_3$ interface, via weak van der Waals interactions, demonstrates excellent stress release capability under thermal and optical aging (**Fig. R4**). This conclusion directly addresses the present question as well. Please refer to our response to **Comment #1** for supporting experimental data and a more comprehensive discussion.

Comment #10. Related to 9, the device stability under various stresses should also be addressed. At least, for the single junction devices.

Response: Thank you for the valuable comments. As detailed in our response to **Comment #1**, we performed separate light (under a 100 mW/cm² white LED) and thermal aging (at 65 °C) tests for encapsulated SnO_x and Sb₂O₃-based PSTs, respectively. Both PSTs retained over 90% of their initial *PCE* after 1,000 hours of continuous aging. This conclusion directly addresses the present question as well. Please refer to our response to **Comment #1** for supporting experimental data and a more comprehensive discussion.

Comment #11. The mechanical toughness of the developed interface remains unclear. Additional analysis is needed to evaluate its resistance to delamination or mechanical stress, particularly in the context of scalable device fabrication and long-term operation.

Response: Thank you for the valuable comments. The connection between C_{60} and Sb_2O_3 is supported by weak van der Waals forces, which possibly results in their low adhesion and mechanical toughness. To further assess the interfacial mechanical toughness of C_{60}/Sb_2O_3 , we deposited Sb_2O_3 films with thicknesses of 40 nm and 300 nm on flexible polyethylene terephthalate (PET)/ C_{60} substrates, respectively. After 500 bending cycles, the 300 nm Sb_2O_3 film developed severe cracking and delamination in the central bending region, whereas the 40 nm Sb_2O_3 film remained intact (**Fig. R41**). Also, the GIXRD measurements of the 40 nm bending sample detected the diffraction peak of the PET substrate at a $\theta_{inc}=3^{\circ}$ (**Fig. R42**). Critically, the XRD peak positions of the C_{60}/Sb_2O_3 showed no shift with varying θ_{inc} , implying minimal stress residual for the 40 nm Sb_2O_3 . Thus, the thinner Sb_2O_3 is well-suited for future application in large-area and flexible photovoltaic devices.

Fig. R41 Photograph and Optical microscopy of the PET/ C_{60} /40 or 300 nm Sb_2O_3 samples before and after 500

cyclic bending tests.

Fig. R42 GIXRD patterns of PET/C₆₀/40 nm Sb₂O₃ sample after 500 cyclic bending measurement at different incidence angles ($\theta_{inc}=0.3, 1$ and 3°).

References

- 8 Zheng, L. et al. Strain-induced rubidium incorporation into wide-bandgap perovskites reduces photovoltage loss. *Science* 388, 88-95 (2025). <https://doi.org/10.1126/science.adt3417>

Comment #12. The manuscript lacks a detailed comparison with other potential buffer layer materials that are frequently used. A discussion on how Sb₂O₃ performs relative to these commonly used alternatives would strengthen the manuscript.

Response: Thank you for the valuable comments. Despite the prevalence of ALD-SnO_x with favorable optoelectronic properties as a buffer layer in high-performance inverted tandem devices^{16,18}, our work highlights the remarkable competitiveness of thermally evaporated Sb₂O₃. As shown in **Fig. 2c** and **Fig. 3d**, the single- and double- junction devices all proved the slightly better performance compared with that of devices based on ALD-SnO_x buffer layer. Specifically, Sb₂O₃ demonstrates comparable electrical characteristics and stability, and even greater advantages in cost-loss and optical performance, establishing it as a compelling alternative.

Besides, thermally evaporated bathocuproine (BCP) is also commonly employed as a buffer layer in single-junction devices^{27,29}. Therefore, we further fabricated single-junction devices based on BCP. We present the *J-V* curve of the champion device, with an excellent *PCE* above 22% (**Fig. R40**). However, previous researches have shown that the BCP, an organic material, generally exhibits poor thermal stability⁶. We also conducted thermal stability tests aging at 85°C for 24 hours, the device based on BCP decomposed macroscopically (**Fig. R43**). Additionally, BCP with poor sputtering resistance cannot be directly used as a buffer layer in tandem devices¹⁴. It needs to be combined with ultra-thin metal (such as BCP: Ag) to use^{30,31}. However, we used BCP: Ag as the buffer layer to prepare tandem solar cells and obtain the champion *PCE* below 23% (**Fig. R39**), possibly attributed to a narrow process window for the fabrication of ultra-thin metals.

Compared with BCP, Sb₂O₃ offers a broader processing window and superior intrinsic stability, making it more suitable as buffer layer for fabricating high-performance large-area tandem solar cells.

Thermal aging at 85°C in N₂ for 24h.

BCP-based PSC Sb₂O₃-based PSC

Fig. R43 Photographs of BCP- and Sb₂O₃-based devices under thermal aging at 85°C in a N₂ atmosphere.

References

- 7 Chen, P. *et al.* Multifunctional ytterbium oxide buffer for perovskite solar cells. *Nature* **625**, 516-522 (2024). <https://doi.org/10.1038/s41586-023-06892-x>
- 14 Liu, K. *et al.* Reducing sputter induced stress and damage for efficient perovskite/silicon tandem solar cells. *Journal of Materials Chemistry A* **10**, 1343-1349 (2022). <https://doi.org/10.1039/d1ta09143c>
- 16 Jia, L. *et al.* Efficient perovskite/silicon tandem with asymmetric self-assembly molecule. *Nature* **644**, 912-919 (2025). <https://doi.org/10.1038/s41586-025-09333-z>
- 18 Wu, W. *et al.* Stable and uniform self-assembled organic diradical molecules for perovskite photovoltaics. *Science* **387**, eadv4551 (2025). <https://doi.org/10.1126/science.adv4551>
- 27 Park, S. M. *et al.* Low-loss contacts on textured substrates for inverted perovskite solar cells. *Nature* **624**, 289-294 (2023). <https://doi.org/10.1038/s41586-023-06745-7>
- 29 Qu, G. *et al.* Self-assembled materials with an ordered hydrophilic bilayer for high performance inverted Perovskite solar cells. *Nature Communications* **16** (2025). <https://doi.org/10.1038/s41467-024-55523-0>
- 30 Zheng, J. *et al.* Polycrystalline silicon tunnelling recombination layers for high-efficiency perovskite/tunnel oxide passivating contact tandem solar cells. *Nature Energy* **8**, 1250-1261 (2023). <https://doi.org/10.1038/s41560-023-01382-w>
- 31 Wang, X. *et al.* Long-chain anionic surfactants enabling stable perovskite/silicon tandems with greatly suppressed stress corrosion. *Nature Communications* **14**, 2166 (2023). <https://doi.org/10.1038/s41467-023-37877-z>

Comment #13. A thorough evaluation, supported by quantitative data, is necessary to assess the economic and technical feasibility of implementing the Sb_2O_3 interlayer in large-scale manufacturing, especially in comparison with commonly adopted strategies.

Response: Thank you for the valuable comments. SnO_x is currently the most commonly used buffer layer in perovskite/silicon tandem solar cells. Replacing SnO_x with Sb_2O_3 results in higher short-circuit current density and power conversion efficiency in tandem solar cells, and the parameters in the main text demonstrate the technical feasibility of using Sb_2O_3 as a buffer layer. Taking the highest-efficiency cell in the text as an example, the conventional combination is 15 nm C_{60} /15 nm SnO_x , while the innovative combination in this work is 5 nm C_{60} /15 nm Sb_2O_3 .

Currently, the techno-economic analysis of tandem cells uses a bottom-up model to predict the levelized cost of electricity (LCOE)³²⁻³⁴. The cost of each process step includes materials, equipment, labor, utilities, maintenance, building and facilities costs. However, this analysis method is highly influenced by factors such as region, policies, and production line capacity, making the model complex and subject to continuous updates as production line designs evolve³⁴. Given that the tandem cell in this study only modified the C_{60} and buffer layer, we calculated the material cost, equipment depreciation cost, electricity cost, and sum of them for the different combination of C_{60} and buffer layer on a 100 cm² silicon substrate in our laboratory (**Table S8**). The total cost of 15 nm C_{60} /15 nm SnO_x is about 3.4 times that of 5 nm C_{60} /15 nm Sb_2O_3 for each experiment, demonstrating the economic feasibility of using Sb_2O_3 as a buffer layer. (Note that material utilization factor is closely related to factors such as equipment type, geometry, and deposition process. In this statistics, thin films were deposited according to steps in the experimental procedure, and the material loss for each experiment was determined by weighing the source material before and after deposition using a balance. The average material utilization factor was obtained through film quality divided by material loss from three different film thickness

experiments. ^bNote that the unit prices of materials and equipment investments are based on the costs of the products and equipment used in our laboratory. ^cNote that the processing duration of evaporated film and ALD-deposited SnO_x is the minimum total time of evacuating the chamber, source heating, stabilizing evaporation rate and depositing film.)

Table R1 Fabrication cost comparison between C₆₀-15/SnO_x and C₆₀-5/Sb₂O₃ stacks for 100 cm² PSTs.

	Common combination		Innovation combination	
	C ₆₀	SnO _x	C ₆₀	Sb ₂ O ₃
Layer thickness per device (nm)	15	15	5	15
Film volume for 100 cm ² (cm ³)	0.00015	0.00015	0.00005	0.00015
Film density (g/cm ³)	1.65 ³⁵	6.90 ³⁶	1.65	5.7 ³⁷
Film quality (g)	0.0002475	0.001035	0.0000825	0.000855
Average material utilization factor ^a	0.1	0.04	0.1	0.06
Usage of precursor (g)	0.002475	0.025875	0.000825	0.01425
Unit price of material ^b (CNY/g)	800	100	800	41 ³⁸
Material Cost (RMB/100 cm ²)	1.98	2.5875	0.66	0.58425
Total Material Cost (RMB/100 cm²)		4.5675		1.24425
Equipment investment (CNY) ^b	394000	360000	394000	
Equipment lifetime (h)	72000	72000	72000	
Equipment depreciation cost (CNY/h)	5.5	5	5.5	
Processing duration (h) ^c	1	2.5	1.1	
Equipment depreciation cost (CNY/100 cm ²)	5.5	12.5	6.05	
Total equipment depreciation cost (CNY/100 cm²)		18		6.05
Equipment power (KW)	6	8	6	
Processing duration (h)	1	2.5	1.1	
Average electricity price (CNY/KW/h)	0.52	0.52	0.52	
Electricity cost (CNY/100 cm ²)	3.12	10.4	3.432	
Total electricity cost		13.52		3.432
Total cost (CNY/100 cm²)		36.09		10.73

References

- 32 Zafoschnig, L. A., Nold, S. & Goldschmidt, J. C. The race for lowest costs of electricity production: techno-economic analysis of silicon, perovskite and tandem solar cells. *IEEE Journal of Photovoltaics* **10**, 1632-1641 (2020). <https://doi.org/10.1109/jphotov.2020.3024739>
- 33 Chang, N. L. *et al.* A bottom-up cost analysis of silicon–perovskite tandem photovoltaics. *Progress in Photovoltaics: Research and Applications* **29**, 401-413 (2020). <https://doi.org/10.1002/pip.3354>
- 34 Cordell, J. J., Woodhouse, M. & Warren, E. L. Technoeconomic analysis of perovskite/silicon tandem solar modules. *Joule* **9** (2025). <https://doi.org/10.1016/j.joule.2024.10.013>
- 35 MatWeb. Carbon (fullerene-C₆₀). <https://matweb.com/search/DataSheet.aspx?MatGUID=079e7b90a5914e24bc272fbb0e15fa9b&ckck=1> (2025).
- 36 MatWeb. Tin oxide nanopowder. <https://matweb.com/search/DataSheet.aspx?MatGUID=8161a17bd90f4ec2b2a16fe15db6c056> (2025).
- 37 MatWeb. Antimony oxide, Sb₂O₃ (Valentinite). <https://matweb.com/search/DataSheet.aspx?MatGUID=7fd3759d5bed4c5b9b832ef8faff6dad> (2025).
- 38 Sigma-Aldrich. Antimony oxide (Cat. No. 379255). <https://www.sigmaaldrich.cn/CN/zh/product/aldrich/379255> (2025).

Comment #14. For the certification of the large-area tandem device, data on hysteresis behavior and steady-state power output are missing. These measurements are essential to validate the reliability and performance consistency of the device.

Response: Thank you for the valuable comments. We have provided the forward and reverse $J-V$ scans of the encapsulated large-area champion perovskite/silicon tandem solar cell (64.64 cm^2) to illustrate its hysteresis behavior in Fig. R44 (Fig. 4d in Manuscript). According to your suggestions, the steady-state power output (SPO) of 27.66% was also recorded after 1800 s MPP tracking, as shown in Fig. R45 (Fig. S45 in SI).

Fig. R44 The $J-V$ curves of the encapsulated large-area champion SnO_x - and Sb_2O_3 -based PSTs (64.64 cm^2).

Fig. R45 SPO of the 64.64 cm^2 champion PST.

Comment #15. The device performance results using different bandgap perovskites primarily serve to demonstrate generality and would be more appropriate in the Supplementary Information. In contrast, the comparison of electronic properties should be presented in the main manuscript to support the core scientific discussion.

Response: Thank you for the valuable comments. Following your suggestions, we now focus on the comparison of electronic properties between SnO_x- and Sb₂O₃-based devices to support our scientific viewpoints.

As shown in Fig. R46 (Fig. 2g in Manuscript), TPC measurements of SnO_x- and Sb₂O₃-based devices reveal comparable photocurrent decay lifetimes, indicating that Sb₂O₃-based devices achieve efficient charge extraction for the unique defect-assisted transport mode of Sb₂O₃. In Fig. R47 (Fig. 2h in Manuscript), the similar photo voltage decay lifetimes observed in TPV measurements confirm that the gap states in Sb₂O₃ do not introduce significant non-radiative recombination. Collectively, these results provide strong evidence for the effectiveness of the defect-mediated conduction mechanism in Sb₂O₃.

Fig. R46 TPC measurements of SnO_x- and Sb₂O₃-based PSCs.

Fig. R47 TPV measurements of SnO_x- and Sb₂O₃-based PSCs.

Comment #16. The light stability results show no significant difference between the SnO₂- and Sb₂O₃-based tandem devices. If SnO₂ were causing substantial damage to the perovskite surface, a corresponding impact on device stability would be expected. This apparent inconsistency should be addressed and clarified.

Response: Thank you for the valuable comments. Based on XPS results (Fig. R48), the deposition of ALD-SnO_x triggers an irreversible interfacial reaction with the perovskite. Thus, we agree with that if SnO_x directly deposited on the surface of perovskite, a significant impact on stability would be expected. However, the C₆₀ layer serves to block this detrimental contact while facilitating electron transport. We found that the detrimental effects of ALD-SnO_x are progressively mitigated with thicker C₆₀ layers, with normal device performance recovery achieved beyond 15 nm (Fig. R49). In previous extensive works, the thicker C₆₀ layers are commonly used in tandem devices with SnO_x to ensure superior performance and stability. Indeed, our ALD-SnO_x-based devices with a 15 nm C₆₀ layer exhibited excellent stability in aging tests.

Fig. R48 XPS spectra of core levels (N 1s, I 3d, Pb 4f, Sb 3d, Sn 3d) for the pristine PVK and PVK/SnO_x or Sb₂O₃.

Fig. R49 Statistics of *PCE* of the PSCs with different C₆₀ thicknesses based on SnO_x and Sb₂O₃, respectively.

References

- 1 Palmstrom, A. F. *et al.* Interfacial effects of tin oxide atomic layer deposition in metal halide perovskite photovoltaics. *Advanced Energy Materials* **8**, 1800591 (2018). <https://doi.org/10.1002/aenm.201800591>
- 2 Zhou, B., Zhou, W. & Wu, P. Ferromagnetic ordering and metallic-like conductivity in sputtered SnNx films. *Journal of Alloys and Compounds* **604**, 106-111 (2014). <https://doi.org/10.1016/j.jallcom.2014.03.098>
- 3 Hultqvist, A. *et al.* SnO_x atomic layer deposition on bare perovskite—an investigation of initial growth dynamics, interface chemistry, and solar cell performance. *ACS Applied Energy Materials* **4**, 510-522 (2021). <https://doi.org/10.1021/acsaem.0c02405>
- 4 Liu, K. *et al.* A wafer-scale van der Waals dielectric made from an inorganic molecular crystal film. *Nature Electronics* **4**, 906-913 (2021). <https://doi.org/10.1038/s41928-021-00683-w>
- 5 Liu, J. *et al.* Efficient and stable perovskite-silicon tandem solar cells through contact displacement by MgFx. *Science* **377**, 302-306 (2022). <https://doi.org/10.1126/science.abn8910>
- 6 Chen, P. *et al.* Multifunctional ytterbium oxide buffer for perovskite solar cells. *Nature* **625**, 516-522 (2024). <https://doi.org/10.1038/s41586-023-06892-x>
- 7 Linarez Pérez, O. E., Sánchez, M. D. & López Teijelo, M. Characterization of growth of anodic antimony oxide films by ellipsometry and XPS. *Journal of Electroanalytical Chemistry* **645**, 143-148 (2010). <https://doi.org/10.1016/j.jelechem.2010.04.023>
- 8 Zheng, L. *et al.* Strain-induced rubidium incorporation into wide-bandgap perovskites reduces photovoltage loss. *Science* **388**, 88-95 (2025). <https://doi.org/10.1126/science.adt3417>
- 9 Han, W. *et al.* Two-dimensional inorganic molecular crystals. *Nature Communications* **10**, 4728 (2019). <https://doi.org/10.1038/s41467-019-12569-9>
- 10 Tigau, N. *et al.* The influence of heat treatment on the electrical conductivity of antimony trioxide thin films. *Journal of Optoelectronics and Advanced Materials* **5**, 907-912 (2003).
- 11 Yang, Q. *et al.* Origin of sputter damage during transparent conductive oxide deposition for semitransparent perovskite solar cells. *Journal of Materials Chemistry A* **12**, 14816-14827 (2024). <https://doi.org/10.1039/d3ta06654a>
- 12 Kanda, H. *et al.* Analysis of sputtering damage on I–V curves for perovskite solar cells and simulation with reversed diode model. *The Journal of Physical Chemistry C* **120**, 28441-28447 (2016). <https://doi.org/10.1021/acs.jpcc.6b09219>
- 13 Härtel, M. *et al.* Reducing sputter damage-induced recombination losses during deposition of the transparent front-electrode for monolithic perovskite/silicon tandem solar cells. *Solar Energy Materials and Solar Cells* **252**, 112180 (2023). <https://doi.org/10.1016/j.solmat.2023.112180>
- 14 Liu, K. *et al.* Reducing sputter induced stress and damage for efficient perovskite/silicon tandem solar cells. *Journal of Materials Chemistry A* **10**, 1343-1349 (2022). <https://doi.org/10.1039/d1ta09143c>
- 15 Magliano, E. *et al.* Solution-processed metal-oxide nanoparticles to prevent the sputtering damage in perovskite/silicon tandem solar cells. *ACS Applied Materials & Interfaces* **17**, 17599-17610 (2025). <https://doi.org/10.1021/acsaami.5c00090>
- 16 Jia, L. *et al.* Efficient perovskite/silicon tandem with asymmetric self-assembly molecule. *Nature* **644**, 912-919 (2025). <https://doi.org/10.1038/s41586-025-09333-z>
- 17 Liu, J. *et al.* Perovskite/silicon tandem solar cells with bilayer interface passivation. *Nature* **635**, 596-603 (2024). <https://doi.org/10.1038/s41586-024-07997-7>
- 18 Wu, W. *et al.* Stable and uniform self-assembled organic diradical molecules for perovskite photovoltaics. *Science* **387**, eadv4551 (2025). <https://doi.org/10.1126/science.adv4551>

- 19 Chin, X. Y. *et al.* Interface passivation for 31.25%-efficient perovskite/silicon tandem solar cells. *Science* **381**, 59-62 (2023). <https://doi.org/10.1126/science.adg0091>
- 20 Er-Raji, O. *et al.* Electron accumulation across the perovskite layer enhances tandem solar cells with textured silicon. *Science*, eadx1745 (2025). <https://doi.org/10.1126/science.adx1745>
- 21 Li, Y. *et al.* CsCl induced efficient fully-textured perovskite/crystalline silicon tandem solar cell. *Nano Energy* **122**, 109285 (2024). <https://doi.org/10.1016/j.nanoen.2024.109285>
- 22 Xu, Q. *et al.* Conductive passivator for efficient monolithic perovskite/silicon tandem solar cell on commercially textured silicon. *Advanced Energy Materials* **12**, 2202404 (2022). <https://doi.org/10.1002/aenm.202202404>
- 23 Xu, Q. *et al.* Diffusible capping layer enabled homogeneous crystallization and component distribution of hybrid sequential deposited perovskite. *Advanced Materials* **36**, 2308692 (2024). <https://doi.org/10.1002/adma.202308692>
- 24 Liu, J. *et al.* Textured perovskite/silicon tandem solar cells achieving over 30% efficiency promoted by 4-fluorobenzylamine hydroiodide. *Nano-Micro Letters* **16**, 189 (2024). <https://doi.org/10.1007/s40820-024-01406-4>
- 25 Xu, L. *et al.* Accurate optical modeling of monolithic perovskite/silicon tandem solar cells and modules on textured silicon substrates. *PRX Energy* **1**, 023005 (2022). <https://doi.org/10.1103/PRXEnergy.1.023005>
- 26 Zhang, X. *et al.* A spiro-type self-assembled hole transporting monolayer for highly efficient and stable inverted perovskite solar cells and modules. *Energy & Environmental Science* **18**, 468-477 (2025). <https://doi.org/10.1039/d4ee01960a>
- 27 Park, S. M. *et al.* Low-loss contacts on textured substrates for inverted perovskite solar cells. *Nature* **624**, 289-294 (2023). <https://doi.org/10.1038/s41586-023-06745-7>
- 28 Zheng, X. *et al.* Co-deposition of hole-selective contact and absorber for improving the processability of perovskite solar cells. *Nature Energy* **8**, 462-472 (2023). <https://doi.org/10.1038/s41560-023-01227-6>
- 29 Qu, G. *et al.* Self-assembled materials with an ordered hydrophilic bilayer for high performance inverted Perovskite solar cells. *Nature Communications* **16** (2025). <https://doi.org/10.1038/s41467-024-55523-0>
- 30 Zheng, J. *et al.* Polycrystalline silicon tunnelling recombination layers for high-efficiency perovskite/tunnel oxide passivating contact tandem solar cells. *Nature Energy* **8**, 1250-1261 (2023). <https://doi.org/10.1038/s41560-023-01382-w>
- 31 Wang, X. *et al.* Long-chain anionic surfactants enabling stable perovskite/silicon tandems with greatly suppressed stress corrosion. *Nature Communications* **14**, 2166 (2023). <https://doi.org/10.1038/s41467-023-37877-z>
- 32 Zafoschnig, L. A., Nold, S. & Goldschmidt, J. C. The race for lowest costs of electricity production: techno-economic analysis of silicon, perovskite and tandem solar cells. *IEEE Journal of Photovoltaics* **10**, 1632-1641 (2020). <https://doi.org/10.1109/jphotov.2020.3024739>
- 33 Chang, N. L. *et al.* A bottom-up cost analysis of silicon–perovskite tandem photovoltaics. *Progress in Photovoltaics: Research and Applications* **29**, 401-413 (2020). <https://doi.org/10.1002/pip.3354>
- 34 Cordell, J. J., Woodhouse, M. & Warren, E. L. Technoeconomic analysis of perovskite/silicon tandem solar modules. *Joule* **9** (2025). <https://doi.org/10.1016/j.joule.2024.10.013>
- 35 MatWeb. Carbon (fullerene-C₆₀). <https://matweb.com/search/DataSheet.aspx?MatGUID=079e7b90a5914e24bc272fbb0e15fa9b&ckck=1> (2025).
- 36 MatWeb. Tin oxide nanopowder. <https://matweb.com/search/DataSheet.aspx?MatGUID=8161a17bd90f4ec2b2a16fe15db6c056> (2025).
- 37 MatWeb. Antimony oxide, Sb₂O₃ (Valentinite). <https://matweb.com/search/DataSheet.aspx?MatGUID=7fd3759d5bed4c5b9b832ef8faff6dad> (2025).

38 Sigma-Aldrich. Antimony oxide (Cat. No. 379255). <https://www.sigmaaldrich.cn/CN/zh/product/aldrich/379255> (2025).

Reviewer #3 (Remarks to the Author):

The manuscript has been improved significantly. However, several important issues remain insufficiently addressed, particularly regarding the tandem characterization, e.g., the textured Si bottom cell, and the functional layers in the full stack.

1. In this work, long-term stability and MPP tracking are performed under a 100 mW cm^{-2} white LED source, for single-junction and especially for tandem devices. While this is experimentally convenient, it raises serious concerns for monolithic perovskite/Si tandems, which is not acceptable.

Simply, the spectral distribution of a “white LED” typically differs from AM 1.5G, particularly in the blue and near-IR regions that are critical for current matching between the perovskite top and Si bottom cells, and bias subcells to affect its efficiency decay dynamics. I therefore recommend that the authors to Demonstrate stability under AM 1.5G. Without such analysis, the reported “negligible efficiency loss” under LED illumination cannot be directly interpreted as stability under realistic PV operating conditions.

Response: Thank you for the valuable comments. We fully understand and acknowledge the concern that the spectral mismatch between a white LED and AM 1.5G could influence the precise assessment of the current-matching and the degradation dynamics in perovskite/Si tandem devices.

However, we pioneered a novel functional material in this study. For an initial stability benchmark, we used a white LED as a relative proof-of-concept, following the precedent of

recent high-quality literature¹⁻⁴.

References

- 1 Ding, Z. *et al.* Highly passivated TOPCon bottom cells for perovskite/silicon tandem solar cells. *Nature Communications* **15** (2024). <https://doi.org/10.1038/s41467-024-52309-2>
- 2 Wu, W. *et al.* Stable and uniform self-assembled organic diradical molecules for perovskite photovoltaics. *Science*, eadv4551 (2025).
- 3 Sun, Y. *et al.* Flexible perovskite/silicon monolithic tandem solar cells approaching 30% efficiency. *Nature Communications* **16** (2025). <https://doi.org/10.1038/s41467-025-61081-w>
- 4 Liu, N. *et al.* Particle decoration enables solution-processed perovskite integration with fully-textured silicon for efficient tandem solar cells. *Nature Communications* **16** (2025). <https://doi.org/10.1038/s41467-025-64546-0>

Nevertheless, the results of device stability observed under LED illumination cannot be directly interpreted as stability under realistic PV operating conditions. Therefore, regarding all the description of device stability, we have all specified the LED lighting condition and made corresponding modifications in the text.

A full investigation under standard AM 1.5G conditions can be considered to furtherly advance this pioneering work in the future.

2. The tandem devices are based on double-textured SHJ bottom cells with “sub-micron-pyramid-structured” surfaces. However, the manuscript does not provide quantitative information on this texture, e.g., pyramid height, base width, pitch, or RMS roughness. Moreover, the etching process (e.g., alkaline vs acidic texturing, etch time, orientation dependence) is not described in enough detail to allow reproducibility.

Response: Thank you for the valuable comments. The double-textured silicon heterojunction bottom cells used in this study are commercial products provided by LONGi Green Energy Technology Co., Ltd. Both sides of the substrate were mildly textured using a relatively low-concentration alkaline solution to form small-sized pyramid structures. This process is consistent in principle with the front-side process described in LONGi’s published literature⁵. The fabrication process of silicon bottom cells has been added into support information in detail and marked with color.

We also have performed direct characterization on the substrates used in this work via SEM (**Fig. R1**), which shows that the front-side and rear-side pyramid size is about 300-500 nm.

Fig. R1 Cross-sectional SEM images of the double-textured SHJ bottom cell.

- 5 Liu, J. *et al.* Perovskite/silicon tandem solar cells with bilayer interface passivation. *Nature* **635**, 596-603 (2024). <https://doi.org/10.1038/s41586-024-07997-7>
-

3. Most of the detailed structural and electronic characterization of Sb₂O₃ (AFM, TEM, GIXRD, c-AFM, UPS, defect-state analysis) is performed on planar substrates or planar stacks. In contrast, the actual tandem devices use Sb₂O₃ deposited on a NiO_x/ITO-coated, pyramid-textured SHJ surface, followed by sputtered IZO and Ag or screen-printed contacts. I suggest the authors providing detailed characterizations on the actual textured tandem stack along and across pyramids, showing continuity and thickness of Sb₂O₃ over the entire facet and its crystallinity (nanocrystalline as claimed in the manuscript).

Response: Thank you for the valuable comments. First, it is important to clarify that one of the primary objectives of this study is to investigate the intrinsic material properties of evaporated Sb₂O₃ as a novel buffer layer, such as its micro structure, electronic properties, and defect states. Therefore, conducting systematic physical characterizations on planar substrates or planar stacks is a universal and effective methodology in materials science. This approach allows us to clearly elucidate its core physical mechanisms under controlled conditions, free from interference by complex morphologies.

More importantly, thanks to the submicron-sized pyramids on our substrate, the process ensures that the resulting perovskite layer filled up the underlying pyramid, resulting in a quasi-planar top surface for the perovskite layer. The critical morphological transformation is unambiguously confirmed by cross-sectional TEM and SEM images of the complete tandem device or films (**Fig. R2-3**).

Consequently, the subsequent layers, including the Sb₂O₃ buffer layer, the IZO, and the metal contacts, are deposited on the like-planar perovskite surface. Therefore, the detailed structural and electronic characterizations of Sb₂O₃ that we performed on planar stacks are

almost representative of its state in the actual working tandem device.

Fig. R2 Cross-sectional TEM image and EDX mappings of Si/NiO_x/Me-4PACz/PVK/C₆₀/Sb₂O₃/IZO stack.

Fig. R3 Cross-sectional SEM images of TE-PbI₂, PVK and PVK/C₆₀/Sb₂O₃/IZO stack films deposited on the micro-textured silicon bottom cell.

4. Regarding large-area uniformity, the *PCE* distribution of sufficient numbers of small cells on a 10×10 cm² substrate, are used to support scalability. Please also comment on the yield and variability among large-area devices fabricated under nominally identical conditions.

Response: Thank you for the valuable comments. Regarding large-area uniformity, we have provided the *PCE* distribution of 100 sub-cells fabricated on a 10×10 cm² substrate, which demonstrates good uniformity across the area (**Fig. R4**). Concerning yield and batch-to-batch variability, our study remains at the lab-scale stage, and thus large-volume production has not been carried out. While an exact production yield cannot be provided, we have evaluated the reproducibility by presenting the performance of four large-area tandem devices fabricated in separate batches under the same conditions.

The photovoltaic parameters of these four devices, each representing an independent batch, are summarized in **Fig. R5** and **Table R1**. The devices exhibit a high average *PCE* of 27.82% with an exceptionally low relative standard deviation (RSD) of 1.52% for the *PCE*. The other key parameters also show remarkably low RSDs, all below 2%. Crucially, this excellent consistency was achieved across four separate batches, which powerfully demonstrates the high reproducibility and robustness of our fabrication process. Manuscript and supporting information have been made corresponding modification and marked with color.

Fig. R4 Statistical distribution of normalized *PCE* with 100 sub-cells based on Sb_2O_3 on a $10 \times 10 \text{ cm}^2$ substrate.

Fig. R5 *J-V* curves of four large-area (64.64 cm^2) devices from different fabrication batches.

Table. R1 PV parameters of four large-area (64.64 cm²) devices from different fabrication batches.

	V _{oc} (V)	J _{sc} (mA/cm ²)	FF (%)	PCE (%)
Device 1	1.839	18.93	80.02	27.86
Device 2	1.840	18.95	80.42	28.04
Device 3	1.862	18.66	78.31	27.21
Device 4	1.832	18.96	81.11	28.16
Average	1.843	18.88	79.97	27.82
Std. Dev.	0.0130	0.1439	1.192	0.4233
RSD	0.71%	0.76%	1.49%	1.52%

5. The authors attribute the inferior long-term stability of Sb₂O₃-based PSTs to poorer compactness and facilitated ion migration through the molecular-crystal Sb₂O₃. Please provide direct evidence to support the claim.

Response: Thank you for the valuable comments. To verify our initial speculation regarding the stability difference, we performed XPS analysis on the PVK/C₆₀/SnO_x and PVK/C₆₀/Sb₂O₃ structures before and after photo-thermal aging (**Fig. R6-7**), which yielded key findings. Firstly, regarding ion migration, the iodine signal decreased on the aged SnO_x surface but increased significantly on the aged Sb₂O₃-based surface. This provides direct evidence for the slightly inferior halide ion migration suppression capability of the Sb₂O₃ layer, consistent with its relatively poor compactness. Second, the XPS spectra reveal a distinct consumption of metallic antimony (Sb⁰) in the aged Sb₂O₃ sample. Since conduction mechanism of Sb₂O₃ buffer layer relies on a quasi-defect band induced by antimony interstitial atoms (I_{Sb}), the loss of Sb⁰ implies the degradation of the crucial electron transport pathway. Therefore, the inferior stability of the Sb₂O₃-based device may arise from its weaker barrier against halide ion migration and the consumption of the I_{Sb}.

Fig. R6 XPS spectra (I 3d) of the pristine and light-thermally aged SnO_x -based and Sb_2O_3 -based samples under white LED (100 mW cm^{-2}) at $65 \text{ }^\circ\text{C}$ in a N_2 atmosphere.

Fig. R7 XPS spectra (Sn 3d and Sb 3d) of the pristine and light-thermally aged SnO_x -based and Sb_2O_3 -based samples under white LED (100 mW cm^{-2}) at $65 \text{ }^\circ\text{C}$ in a N_2 atmosphere.

6. Please ensure that all acronyms (TE, PST, PVK, etc.) are defined at first use.

Response: Thank you for the valuable comments. We have revised the manuscript to define all acronyms upon their first use. Specifically, "TE" is now defined as "thermally evaporated", "PST" as "perovskite/silicon tandem", and "PVK" as "perovskite" in the manuscript. All other acronyms have been similarly treated throughout the text.

References

- 1 Ding, Z. *et al.* Highly passivated TOPCon bottom cells for perovskite/silicon tandem solar cells. *Nature Communications* **15** (2024). <https://doi.org/10.1038/s41467-024-52309-2>
- 2 Wu, W. *et al.* Stable and uniform self-assembled organic diradical molecules for perovskite photovoltaics. *Science*, eadv4551 (2025).
- 3 Sun, Y. *et al.* Flexible perovskite/silicon monolithic tandem solar cells approaching 30% efficiency. *Nature Communications* **16** (2025). <https://doi.org/10.1038/s41467-025-61081-w>
- 4 Liu, N. *et al.* Particle decoration enables solution-processed perovskite integration with fully-textured silicon for efficient tandem solar cells. *Nature Communications* **16** (2025). <https://doi.org/10.1038/s41467-025-64546-0>
- 5 Liu, J. *et al.* Perovskite/silicon tandem solar cells with bilayer interface passivation. *Nature* **635**, 596-603 (2024). <https://doi.org/10.1038/s41586-024-07997-7>

Reviewer #3 (Remarks to the Author):

I appreciate the authors' efforts in addressing the majority of the concerns raised in the previous round, particularly regarding the large-area yield statistics and the structural characterization of the silicon bottom cell.

However, the refusal to validate device stability under standard AM 1.5G conditions remains a critical methodological barrier that prevents publication. For monolithic perovskite/silicon tandems, the spectral mismatch of white LEDs fundamentally alters the current-matching condition, forcing sub-cells into artificial bias states that differ significantly from real-world operation. This renders your degradation analysis unreliable, as the perovskite top-cell may not be under the limiting stress encountered in the field. Labeling the work as "pioneering" does not exempt it from basic metrological standards. Without AM 1.5G MPP tracking, I think the claim of device stability is unsubstantiated and the manuscript remains unsuitable for publication.

Response: Thank you for this important comment. We fully understand your concern regarding the potential impact of spectral mismatch (e.g., under white-LED illumination) on stability assessment in monolithic perovskite/silicon tandems, and we agree that stability validation under standard AM 1.5G conditions is an important metrological requirement. To better reflect real-world operation, we followed your suggestion and performed continuous aging tests in air under standard AM 1.5G illumination using a xenon-lamp solar simulator (Model: FG-MN001-ACA-AM1.5G-100) (Figs. R1–R2). The irradiance was calibrated to 1 sun (100 mW/cm²)

using a certified silicon reference cell. The additional AM 1.5G aging test is consistent with our earlier LED aging results (**Fig. R3**), which showed that the Sb_2O_3 -based devices exhibit partly lower stability than the SnO_x -based devices. After 1000 h under LED illumination, the Sb_2O_3 device retained ~90% of its initial PCE, compared with ~97% for SnO_x . In the newly added AM 1.5G test, the corresponding retentions were ~65% (Sb_2O_3) and ~80% (SnO_x) after 450 h. We suggest that the difference in absolute retention between the two tests may be related to the different illumination spectra. The experimental details and corresponding data are provided in the Supplementary Information (**Device Characterization, Supplementary Note 5 and Fig. S39**).

We would like to clarify the scope of this work. The primary contribution is the demonstration of thermally evaporated Sb_2O_3 as a new buffer layer offering advantages in light management over ALD- SnO_x , which translate into improved device efficiency. The stability data are presented as baseline benchmarks to enable objective assessment by the community. As stated in the original submission, the stability of Sb_2O_3 -based devices is not yet on par with the highly mature ALD- SnO_x process, which has benefited from extensive optimization over many years. The newly added AM 1.5G aging results are consistent with the initial stability statement and provide clearer references under standard spectra. We sincerely appreciate your rigorous review, which has improved the scientific rigor of this work.

Fig. R1 Photograph of the stability test for encapsulated SnO_x - and Sb_2O_3 -based devices under continuous AM 1.5G illumination.

Fig. R2 Light aging tests of encapsulated SnO_x - or Sb_2O_3 -based tandem devices under continuous AM 1.5G illumination at $100 \text{ mW}/\text{cm}^2$ in air.

Fig. R3 Light aging tests of encapsulated SnO_x - or Sb_2O_3 -based tandem devices under a white LED lamp illumination at $100 \text{ mW}/\text{cm}^2$ in a N_2 atmosphere.